

# An ensemble estimate of Australian soil organic carbon using machine learning and process-based modelling

Lingfei Wang[1,2], Gab Abramowitz[1,2], Ying-Ping Wang[3], Andy Pitman[1,2] and Raphael A. Viscarra Rossel[4]

[1] ARC Center of Excellence for Climate Extremes, Sydney NSW 2052, Australia

[2] Climate Change Research Center, University of New South Wales, Sydney NSW 2052, Australia

[3] CSIRO Environment, Private Bag 10, Clayton South VIC 3169, Australia

[4] Soil & Landscape Science, School of Molecular & Life Sciences, Faculty of Science & Engineering, Curtin University, GPO Box U1987, Perth WA 6845, Australia.

Correspondence to: Lingfei Wang (lingfei.wang@unsw.edu.au)

## Abstract

Spatially explicit prediction of soil organic carbon (SOC) serves as a crucial foundation for effective land management strategies aimed at mitigating soil degradation and assessing carbon sequestration potential. Here, using more than 1000 in-situ observations, we trained two machine learning models (random forest, and K-means coupled with multiple linear regression), and one process-based model (the vertically resolved MIcrobial-MIneral Carbon Stabilization (MIMICS)) to predict SOC content of the top 30 cm of soil in Australia. Parameters of MIMICS were optimized for different site groupings, using two distinct approaches, plant functional types (MIMICS-PFT), and the most influential environmental factors (MIMICS-ENV). We found that at the continental scale, soil bulk density and mean annual temperature are the dominant controls of SOC variation, and that dominant controls vary for different vegetation types. All models showed good performance in SOC predictions with $R^2$ greater than 0.8 during out-of-sample validation with random forest being the most accurate, and SOC in forests is more predictable than that in non-forest soils. Parameter optimization approaches made a notable difference in the performance of MIMICS SOC prediction with MIMICS-ENV performing better than MIMICS-PFT especially in non-forest soils. Digital maps of terrestrial SOC stocks generated using all the models showed similar spatial distribution with higher values in southeast and southwest Australia, but the magnitude of estimated SOC stocks varied. The mean ensemble estimate of SOC stocks was 30.08 t/ha with K-means coupled with multiple linear regression generating the highest estimate (mean SOC stocks at 38.15 t/ha) and MIMICS-PFT generating the lowest estimate (mean SOC stocks at 24.29 t/ha). We suggest that enhancing process-based models to incorporate newly identified drivers that significantly influence SOC variations in different environments could be key to reducing the discrepancies in these estimates. Our findings underscore the considerable uncertainty in SOC estimates derived from different modelling approaches and emphasize the importance of rigorous out-of-sample validation before applying any one approach in Australia.



## 1. Introduction

Globally, the soil is the largest biogeochemically active terrestrial carbon pool, storing more organic carbon than plants and the atmosphere combined. The turnover of soil organic carbon (SOC) is a key function in plant growth, maintenance of soil water and nutrients, soil structure stabilization and other biogeochemical processes (Lefèvre et al., 2017). Soil can act as either a carbon sink or carbon source depending on the balance of carbon input through plant litter and root exudates and output through respiration and leaching (Terrer et al., 2021; Panchal et al., 2022). Even a small change in SOC stocks, in any direction, could significantly affect the atmospheric concentration of $CO_2$ and thereby climate change (Stockmann et al., 2013).

Given the importance of SOC, there is now a large and growing interest in estimating spatially explicit SOC content and stocks. SOC supports critically important soil-derived ecosystem services, and the amount of SOC indicates the degree of land and soil degradation (Lorenz et al., 2019). SOC content below a certain limit will lead to the decline of microbial diversity, water holding capacity and soil productivity (Stockmann et al., 2015). Additionally, with growing concerns about increasing anthropogenic $CO_2$ emissions, soil carbon sequestration has emerged as a potential strategy for climate change mitigation (Smith, 2016; Rumpel et al., 2018). Protection of existing SOC and rebuilding depleted stocks through land management are potential strategies in mitigating climate change (Bossio et al., 2020). However, effective SOC management requires accurate knowledge of its existing distribution. Reliable estimates of SOC stocks and their spatial variation serve as a reference point for assessing how close soil is to its maximum SOC storage capacity and its potential to sequester additional carbon (Six et al., 2002; Georgiou et al., 2022). Precise estimation of contemporary SOC stocks also provides a baseline map that can be used to calibrate and initialize dynamic-mechanistic models, enabling the study of how SOC will respond to climate and land-use change (Minasny et al., 2013; Viscarra Rossel et al., 2014). It is, for example, a prerequisite for accurately predicting future carbon–climate feedbacks in Earth system models (ESMs) (Todd-Brown et al., 2013).

Accurately assessing SOC storage is challenging due to the complexity of carbon formation and degradation processes in space and time (Keskin et al., 2019). Soil exists as a continuum containing organic compounds at different stages of decomposition (Lehmann and Kleber, 2015). Soil formation can be described by a function of climate, organisms, relief, parent material and time (Jenny, 1994). These factors are widely used in SOC studies for digital soil mapping (McBratney et al., 2003; Viscarra Rossel et al., 2015; Liang et al., 2019). However, the relationship between SOC storage and these driving variables is complex and spatially variable (Mishra and Riley, 2015; Viscarra Rossel et al., 2019; Adhikari et al., 2020) leading to substantial challenges and inherent uncertainties in SOC predictions.

Mechanistic process-based models and empirical models (including machine learning models) are two widely employed approaches used to predict SOC stocks and their spatial distribution. Conventional process-based models assume first-order kinetics for SOC decomposition,




wherein the rate of C decomposition is dependent on temperature and moisture but independent
of microbial biomass, and equilibrium SOC stock is proportional to carbon input and mean
residence time (Abs and Ferrière, 2020; Wang et al., 2021). ESMs coupled with conventional
SOC models cannot accurately predict patterns of contemporary soil carbon and show high
uncertainties in projected SOC dynamics under future climate change (Todd-Brown et al., 2013;
Todd-Brown et al., 2014). This is partly due to the lack of explicit representation of soil
microbial activities and metabolic traits (Wieder et al., 2015). Numerous microbial models have
been developed in the past few decades to improve model performance of SOC predictions
(Chandel et al., 2023), but these models has rarely been incorporated into large-scale modelling
frameworks partly due to the lack of rigorous validation (Luo et al., 2016). Process-based SOC
models are constructed based on our understanding on the major processes governing SOC
dynamics (e.g., carbon input, decomposition, and loss). However, the disagreement in
projections of carbon dynamics by different models highlights the need to improve our
knowledge of SOC cycling (Luo et al., 2016). Machine learning models without any process-
level assumptions provide a tool to identify the most influential controls on SOC variations.
Machine learning models can represent non-linear and non-smooth relationships between
predictor and response variables as well as interactions between different predictors (Heung et
al., 2016). Various machine learning algorithms have been successfully used in digital soil
mapping to predict high-resolution spatially explicit SOC content (Lamichhane et al., 2019).
Several modelling studies of soil carbon content/stocks have been conducted in Australia. Wang
et al. (2018a) trained boosted regression trees and random forest models using observations
from the semi-arid rangelands of eastern Australia. Both models predicted SOC stocks
moderately well based on performance metrics. The fitted models were then applied to map the
spatial distribution of SOC at two soil depths (0-5 cm and 0-30 cm).  Continentally, Viscarra
Rossel et al. (2014) trained the CUBIST model, a form of piecewise linear decision tree, using
more than five thousand observations to produce a high resolution (90 m × 90 m) baseline map
of SOC stocks of Australian terrestrial systems and its uncertainty at 30 cm depth. Based on the
baseline map, Walden et al. (2023) derived spatially explicit estimates of Australian SOC stocks
and uncertainty including additional data from forests from southeastern Australia and coastal
marine (or blue carbon) ecosystems. SOC content at multiple soil depths along with associated
uncertainties were also estimated using different machine learning algorithms (Viscarra Rossel
et al., 2015; Wadoux et al., 2023). Moreover, the distribution of different soil carbon
compositions (i.e., the particulate, mineral-associated and pyrogenic organic carbon fractions)
and the importance of environmental factors on their variations were also studied using machine
learning (Viscarra Rossel et al., 2019). However, despite the progress made in SOC modelling,
significant uncertainties persist in SOC estimates due to the inherent complexities of SOC
variations, the lack of appropriately sampled SOC observations and the amount of data. All
these continental estimates were generated using empirical modelling approaches or first-order
biogeochemical models (Grace et al., 2006; Lee et al., 2021). Estimates from mechanistic SOC
models with explicit representation of microbial metabolism are missing despite offering the



potential to better constrain SOC dynamics under future climate change scenarios in a way that
empirical approaches cannot.

Our primary objective in this paper is to assess the predictability of SOC stocks in Australia.
We generate a range of estimates of terrestrial SOC stocks, employing both process-based and
empirical modelling, and examine why these estimates might differ. First, we discern the
significance of environmental predictors, both at continental and biome scales. We then
evaluate the performance of random forests, k-means with multiple linear regression and the
vertically resolved MIcrobial-MIneral Carbon Stabilization (MIMICS) SOC model with
different parametrization approaches. Finally, we compare the spatial estimates of SOC stocks
using these different approaches across Australia, and discuss their differences and potential
application to future SOC projection.

## 2. Materials and Methods


### 2.1. Model descriptions


#### 2.1.1. Vertically resolved MIMICS



The MIcrobial-MIneral Carbon Stabilization (MIMICS) model (Wieder et al., 2015; Zhang et
al., 2020) is a soil carbon model that explicitly considers relationships between litter quality,
functional trade-offs in microbial physiology, and the physical protection of microbial by-
products in forming stable soil organic matter. There are two litter pools: metabolic ($LIT_m$) and
structural ($LIT_s$) litter (Figure 1), and the partitioning of litter input into metabolic and structural
pools is determined by the chemical properties of the litter. Litter and SOC turnover are
governed by two microbial functional types that exhibit copiotrophic (i.e., r-selected, $MIC_r$) and
oligotrophic (i.e., K-selected, $MIC_k$) growth strategies. The $MIC_r$ is assumed to have higher
growth and turnover rates, and a preference for consuming labile litter ($LIT_m$), while $MIC_k$ is
characterized by lower growth and turnover rates, and a greater competitive advantage when
consuming low-quality litter ($LIT_s$) and chemically recalcitrant SOC. SOC in MIMICS is
divided into three pools: physically protected ($SOC_p$), (bio)chemically recalcitrant ($SOC_c$) and
available ($SOC_a$) carbon (Figure 1).

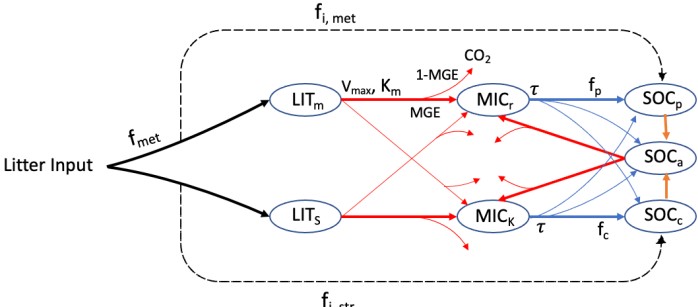

**Figure 1**. Soil carbon pools and fluxes represented in MIMICS (adapted from Wieder et al., (2015)). Litter
inputs are partitioned into metabolic and structural litter pools ($LIT_m$ and $LIT_s$) based on litter quality ($f_{met}$).




Decomposition of litter and available SOC pool ($SOC_a$) are governed by temperature sensitive Michaelis-
Menten kinetics ($V_{max}$ (maximum reaction velocity) and $K_m$ (half saturation constant)), shown by red lines.
Microbial growth efficiency (MGE) determines the partitioning of C fluxes entering microbial biomass pools
vs. heterotrophic respiration. Turnover of microbial biomass ($\tau$, blue) depends on microbial functional types
($MIC_r$ and $MIC_k$), and is partitioned into available, physically protected and chemically recalcitrant SOC
pools ($SOC_a$, $SOC_p$ and $SOC_c$, respectively).

The decomposition of litter pools and SOC pools follows temperature-sensitive Michaelis-
Menten kinetics. Microbial growth efficiency (MGE) determines the partitioning of carbon
fluxes entering microbial biomass pools ($MIC_r$ and $MIC_k$) versus heterotrophic respiration.
Access of microbial enzymes to available substrates is restricted by soil texture. The equations
of MIMCS are from Wieder et al. (2015), except that the density-dependent microbial turnover
was introduced to MIMCS to minimize an unrealistic oscillation (Zhang et al., 2020). To better
simulate carbon turnover at different soil depths, vertical transport of soil carbon was introduced
into MIMICS considering carbon transported through bioturbation and diffusion among
adjacent soil layers (Wang et al., 2021).

Vertically resolved MIMICS is run using a daily time step. The soil was divided into 15 layers,
each of 10 cm thickness. All the sites in this study are assumed to be at steady state (i.e., no
interannual variation of SOC). Historical climate, litterfall input and soil properties were all
assumed to be similar to the average conditions. At each site, the pool size was initialized within
a sensible range for different pools and spun up to finally achieve steady state.

2.1.2.  Machine learning

Two machine learning algorithms were applied in this study to predict SOC. First, random forest
(RF) is a tree-based ensemble learning method that works by building a set of regression trees
and averaging results (Breiman, 2001). Within the training procedure, the RF algorithm
produces multiple trees. Each regression tree in the forest is independently constructed based
on a unique bootstrap sample (with replacement) from the original training data set. The
response, as well as the predictor variables are either categorical (classification trees) or
numeric (regression trees). Bootstrap sampling makes RF less sensitive to overfitting and
allows for robust error estimation based on the remaining test set, the so-called Out-Of-Bag
(OOB) sample (Wiesmeier et al., 2014). We used the "ranger" package R (version 4.2.0) for RF
computation. We trained the RF model with different numbers of trees and observed that the
model's performance remained similar regardless of the number of trees used. The number of
regression trees generated in the forest (num.trees) was finally set as 200, and the number of
predictors randomly selected at each node (mtry) was set as default, which was 2.

Multiple linear regression (MLR) is widely used in SOC studies but found to be less effective
than machine learning algorithms (Lamichhane et al., 2019). Here, instead of applying MLR
directly with all environmental factors as predictors, our approach involved a preliminary step
where we partitioned all observations into distinct clusters using K-means, an unsupervised





machine learning algorithm. K-means aims to segregate the data into a predefined number of
clusters (k), with the objective of maximizing the similarity among data within each cluster.
The underlying assumption here was that sites sharing similar environmental conditions would
exhibit comparable SOC concentration. In cases where certain clusters had fewer observations
than five times the number of predictors, we augmented these clusters by incorporating
observations from other clusters. This augmentation process was guided by the Euclidean
distance between the observation and the cluster centre, ensuring a more robust construction of
the linear regression model. To determine the number of clusters, we applied the coupled K-
means and MLR with varying number of clusters. The selection of the optimal number of
clusters was based on the criterion of producing the smallest root mean square error during
independent out-of-sample validation.

## 2.2. Identification of dominant controllers on SOC concentration

RF-based measures of variable importance have gained widespread popularity as tools for
evaluating the contributions made by predictor variables within a fitted random forest model
(Debeer and Strobl, 2020). In the context of this study, we employed permutation variable
importance (PVI) within the random forest framework to gauge the significance of predictors
in predicting SOC concentration.

The permutation variable importance entails measuring the reduction in a RF model's
performance score upon random shuffling of a single variable values. By doing so, the inherent
relationship between the variable and the SOC concentration is disrupted. Consequently, the
disparity in prediction accuracy observed in a random forest model before and after such
shuffling serves as a quantitative representation of the significance of the particular predictor
in predicting SOC concentration. The greater the importance of the predictor, the higher its
corresponding PVI value becomes.

## 2.3. Parameter optimization

MIMICS parameters were derived from (Zhang et al., 2020; Wang et al., 2021), except that five
parameters (Table 1) which directly control the organic carbon decomposition were optimized.
An effective global optimization algorithm called the shuffled complex evolution (SCE-UA,
version 2.2) method (Duan et al., 1993) was applied for parameter optimization. Parameters
were optimized by minimizing the sum of squared residuals between the observed and modelled
values.

Vertically resolved MIMICS simulated SOC concentration for 15 soil layers. As observations
only provide one measurement for the top 30 cm soil, we computed the average of the modelled
values spanning the 0-10 cm, 10-20 cm, and 20-30 cm soil layers. This average was then
adopted as the modelled SOC concentration for top 30 cm soil, serving as the basis for
evaluating the difference between observations and simulations.



**Table 1.** The optimized model parameters and their value range

| Parameter | Definition | Range |
|---|---|---|
| $a_v$ | A scaling factor for $V_{max}$ | 0-30 |
| $a_k$ | A scaling factor for $K_m$ | 0-20 |
| xdesorp | A scaling factor for SOC desorption rate | 0-3 |
| xbeta | An exponent of the biomass density dependent mortality rate of microbes | 1.05-2 |
| xdiffsoc | A scaling factor for SOC diffusion coefficient in soil | 0-30 |

Parameters were optimized for distinct groups divided based on two approaches. The first
approach involved categorizing all observations into four groups based on plant functional
types (PFTs). The second approach was taking the most influential abiotic variables as
predictors (as outlined in Section 2.2) and dividing all observations into 6 clusters using the K-
means algorithm. The determination of the optimal number of clusters was achieved through
the minimization of the sum of the within-cluster-sum-of-squares-of-all-clusters (WCSSE), a
process facilitated by the "ClusterR" package in R (version 4.2.0). This clustering aimed to
ensure the highest possible similarity among the environmental factors within each cluster. It
was anticipated that SOC ranges within each cluster would be narrow due to the high similarity
of environmental predictors.
## 2.4.    Data
### 2.4.1.    Predictors of SOC concentration
MIMICS requires gridded mean annual temperature (MAT), carbon input and clay content as
driving variables for a spatial simulation. Soil bulk density is also required for conversion
between SOC concentration (g C/kg soil) and SOC stocks (t/ha). Gridded mean annual
precipitation (MAP) and vegetation types were also used during calibration and when
understanding the drivers and spatial variability of SOC. Details of gridded data can be found
in Table 2.
Gridded daily maximum temperature, minimum temperature, and precipitation at 0.05°
resolution were obtained from the SILO database of Australian climate data. Mean daily
temperature was approximated as the average of maximum and minimum daily temperature.
MAT was calculated from mean daily temperature from 1991 to 2020, and MAP was calculated
from daily precipitation from 1991 to 2020.
Carbon input was represented by NPP. Gridded mean annual NPP at 500 m was calculated
based on annual NPP from 2001 to 2020 obtained from MODIS (MOD17A3HGF V6.1). NPP
was partitioned to above-/belowground part by multiplying by the root/shoot ratio for different
vegetation types (Mokany et al., 2006).
The distribution of vegetation types at 3'' resolution was obtained from National Vegetation
Information System (NVIS, version 6.0). Pixels of non-vegetated regions were removed and
types were aggregated to just 4 PFTs: forest, woodland, shrubland and grassland.



Soil bulk density and clay content were obtained from Soil and Landscape Grid National Soil
Attributes Maps (SLGA – Release 2). Soil properties were predicted based on machine learning
at depths 0-5 cm, 5-15 cm, 15-30 cm, 30-60 cm, 60-100 cm, and 100-200 cm in SLGA. Bulk
density and clay content were estimated for top 30 cm soil as weighted average of first 3 layers
in SLGA.

The initial spatial resolution of the gridded data was maintained when extracting the required
environmental factors for each SOC observation. All data were then resampled to 0.05°
resolution using bilinear interpolation for estimation of terrestrial SOC stocks at continental
scale.

**Table 2**. Information of gridded data used in this study.

| | Source | Spatial Scale | Temporal Scale | Unit | Time Period | Ref. |
|---|---|---|---|---|---|---|
| Maximum Temperature | SILO | ~5 km | daily | °C | 1991-2020 | (Jeffrey et al., 2001) |
| Minimum Temperature | SILO | ~5 km | daily | °C | 1991-2020 | (Jeffrey et al., 2001) |
| Precipitation | SILO | ~5 km | daily | mm | 1991-2020 | (Jeffrey et al., 2001) |
| NPP | MODIS | 500 m | annually | g C/m$^2$ | 2001-2020 | (Running and Zhao, 2021) |
| Vegetation Types | NVIS | 100 m | / | / | / | |
| Soil Bulk Density | SLGA | ~90 m | / | kg/m$^3$ | / | (Grundy et al., 2015; Viscarra Rossel et al., 2015) |
| Soil Clay Content | SLGA | ~90 m | / | % | / | (Grundy et al., 2015; Viscarra Rossel et al., 2015) |


2.4.2.  Soil organic carbon observations

SOC observations for top 30 cm soil in Australia were collected from two datasets. The first
one, VR dataset, is described in (Viscarra Rossel et al., 2014; Viscarra Rossel et al., 2019). We
removed the observations collected from croplands based on the land-use record in the dataset
and removed those from unvegetated regions based on NVIS vegetation map (see above).
Observations at 1070 sites remained. SOC stocks were reported in t/ha and converted to SOC
concentration (g C/kg soil) using soil bulk density (BD, kg/m$^3$) and soil depth (m),

$$SOC_{concentration} = SOC_{stock}/(BD \times depth) \times 100 \qquad (1)$$

Clay content and soil bulk density were reported in this dataset and also in Viscarra Rossel et
al. (2015). To better represent SOC distribution in forest, we obtained more forest SOC
observations from a second dataset, the Biomes of Australian Soil Environments (BASE)
described in (Bissett et al., 2016). Here, SOC concentration was reported for 0-10 and 20-30
cm, and we estimated the SOC concentration for 20-30 cm soil using the algorithm (Jobbágy
and Jackson, 2000) below,

$$\log_{10} C = S \log_{10} d + I \qquad (2)$$



Where $C$ represents the SOC stocks (t/ha), and $d$ represents depth. $S$ and $I$ are parameters of the
model. We took the average SOC concentration of three layers as the value for top 30 cm soil.
Clay content was reported in this dataset and bulk density was extracted from SLGA (see
above).
The spatial distribution of SOC concentration observations from different PFTs is shown in
Figure 2a. SOC concentration in top 30 cm is positively skewed, ranging from 1.36 to 59.73 g
C/kg soil with mean value at 9.97 g C/kg soil and median value at 6.11 g C/kg soil. SOC
concentration in grassland, shrubland and woodland show similar distribution patterns (Figure
2b), while SOC concentration in forest is more variable with a standard deviation at 15.92 g
C/kg soil.

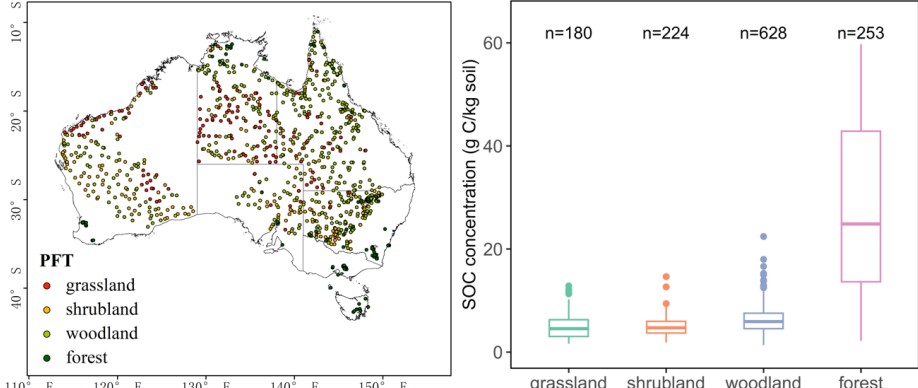

**Figure 2**. Spatial distribution of 1285 soil organic carbon concentration observations used in this study and
the plant functional types which they belong to (a); boxplots of SOC concentration distributions for each
plant functional type (b). For boxplots, centre lines represent the median value, and upper and lower box
boundaries represent third and first quartile. Whiskers extend to the smallest and largest values within 1.5
times the interquartile range.

2.5. Model evaluation

For machine learning models, all observations were separated to a training and test dataset
randomly with 70% used to train the model and the remaining 30% used to validate the
predictions of SOC concentration. For vertically resolved MIMICS, parameters were optimized
for each group, and we again randomly selected 70% of observations in each group to train the
model and used the remaining 30% for validation. To cross-validate, the procedure was repeated
10 times.
The performance of models was evaluated using four metrics. Mean Absolute Error (MAE)
indicates how close the average predictions are to average observations. Root Mean Square
Error (RMSE) measures the overall accuracy combining mean, standard deviation differences





(across sites) and (spatial) correlation. Coefficient of determination ($R^2$) measures the
percentage of variation explained by the model. Lin's Concordance Correlation Coefficient
(LCCC) (Lawrence and Lin, 1989) measures the level of agreement between predictions and
observations following the 1:1 line. A good model will have MAE and RMSE close to 0 and $R^2$
and LCCC close to 1.
2.6.    Estimation of terrestrial SOC stocks
To examine terrestrial SOC stocks and their continental-scale spatial distribution, we generated
pixel-based SOC maps utilizing the four models validated within this study. In the cases of
MIMICS-PFT and MIMICS-ENV, the initial step involved segregating all pixels into four
distinct plant functional groups or six environmental clusters. Since cross-validation was
performed, the machine learning and process-based models were evaluated using test data, and
the models with the optimal performance were subsequently employed at each pixel to estimate
terrestrial SOC stocks. The map of ensemble estimate of SOC stocks was also produced as the
average of four models at each pixel.

## 3. Results

3.1.    Dominant environmental controls of SOC concentration
Using the Permutation Variable Importance (PVI) in random forest, we identified the
significance of environmental factors in predicting SOC. At the continental scale, soil bulk
density is the most influential driver of SOC concentration variations, following by MAT, NPP
and MAP (Figure 3). Soil clay content and plant functional type exhibit relatively lesser
significance in this regard.
The relative predictor importance for forests and grasslands aligns with the importance at
continental scale. In shrubland and woodland, NPP and MAP emerge as the pivotal factors.
Collectively, across both continental and regional scales, soil bulk density, MAT, and MAP are
the three most influential abiotic factors.

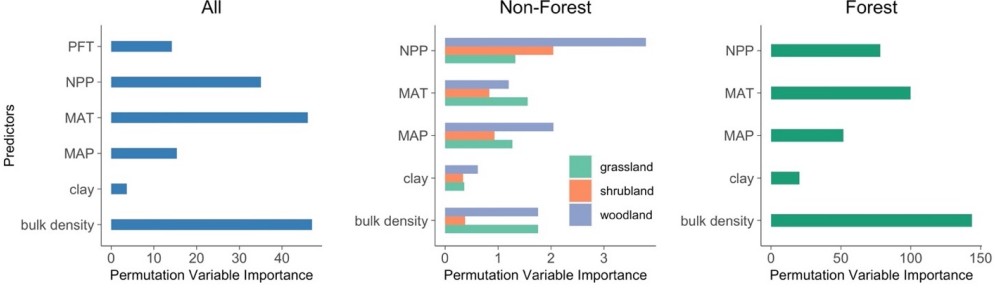

**Figure 3**. Importance of predictors on SOC concentration for different plant functional types.



3.2.    Data clustering based on environmental factors

To develop the calibration groups for MIMICS-ENV, we partitioned the top three important
abiotic factors, which are soil bulk density, MAT and MAP, into six distinct clusters using K-
means (see Section 2.3). The resulting characteristics and SOC distributions for these six
clusters are illustrated in Figure 4.

Notably, a substantial majority of forests were assigned to clusters 2 and 6 (Figure 4a), while
woodland, shrubland, and grassland observations were distributed across the remaining four
clusters. Among these clusters, cluster 5 exhibits the lowest SOC concentration, while SOC of
cluster 1 and 3 display a comparable pattern but spread across different biomes. Conversely,
distribution of SOC concentration in clusters 2, 4, and 6 shows more pronounced variability
(Figure 4b).

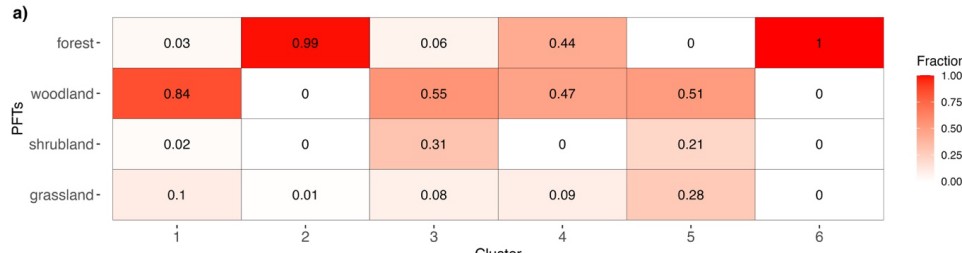

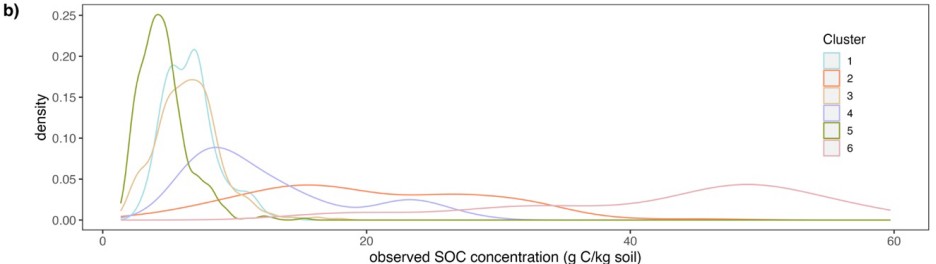

**Figure 4**. Fraction of different PFTs in each cluster divided based on environmental factors (a) and density
plot of observed SOC concentration for different clusters (b).

3.3.    Evaluation of model performance

All models employed in this study (RF, K-means + MLR, MIMICS-PFT and MIMICS-ENV)
predicted SOC concentration well for both training data and test data (Figure 5). As anticipated,
performance for both process-based model and machine learning models degrade using out-of-
sample data versus in-sample training or calibration data. When using test data, the mean value
of $R^2$ for all models ranges from 0.82 to 0.94, mean LCCC ranges from 0.90 to 0.97, mean
RMSE ranges from 2.88 to 4.51 g C/kg soil, and mean MAE ranges from 1.55 to 2.57 g C/kg
soil.





The machine learning models outperformed MIMICS in predicting SOC concentration,
regardless of the optimisation approach taken. Particularly, the random forest algorithm
demonstrated the most accurate predictions characterized by higher $R^2$ and LCCC values and
lower RMSE and MAE values for both training and test data. While MIMICS-ENV displayed
performance similar to that of MIMICS-PFT in SOC concentration predictions based on RMSE
and MAE, the former exhibited slightly superior median $R^2$ and LCCC values (Figure 5).

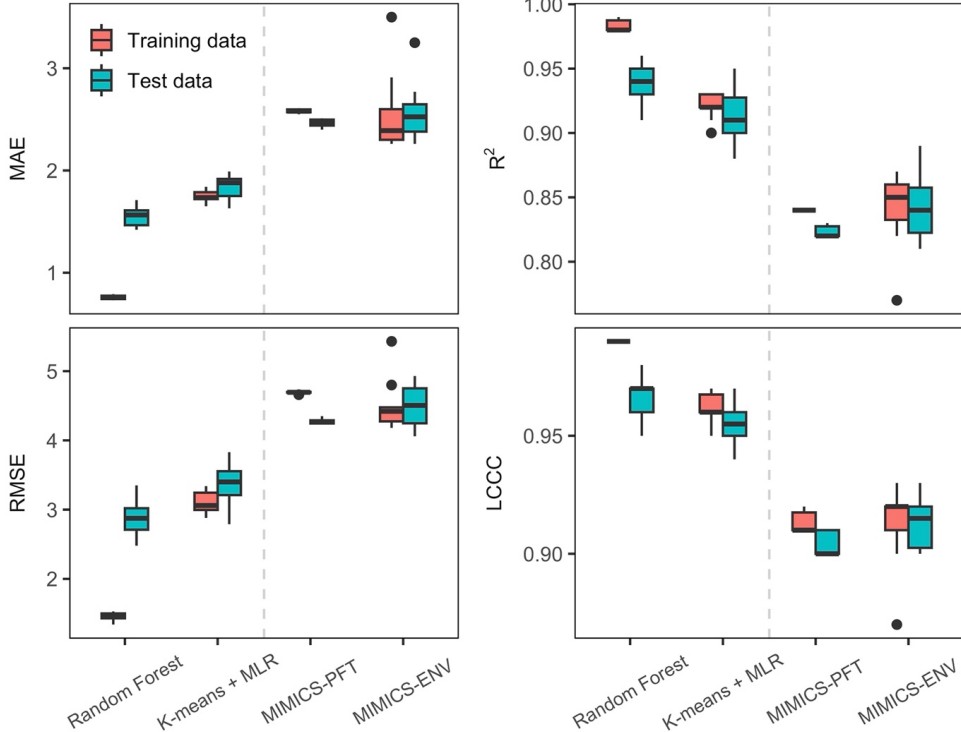

**Figure 5**. Performance metrics of SOC concentration predictions. Units for MAE and RMSE are g C/kg.
Centre line represents median value, and upper and lower box boundaries represent third and first quartile.
Whiskers extend to the smallest and largest values within 1.5 times the interquartile range.
SOC concentration in forest soil exhibited significantly higher predictability than those in non-
forest (woodland, shrubland and grassland) soil, evidenced by higher $R^2$ (ranging from 0.58 to
0.91) and LCCC (ranging from 0.75 to 0.95) for test data (Figure 6). Machine learning
algorithms surpassed MIMICS in predicting SOC for both forest and non-forest soils. Notably,
MIMICS-ENV outperformed MIMICS-PFT in SOC predictions, particularly in non-forest
soils.

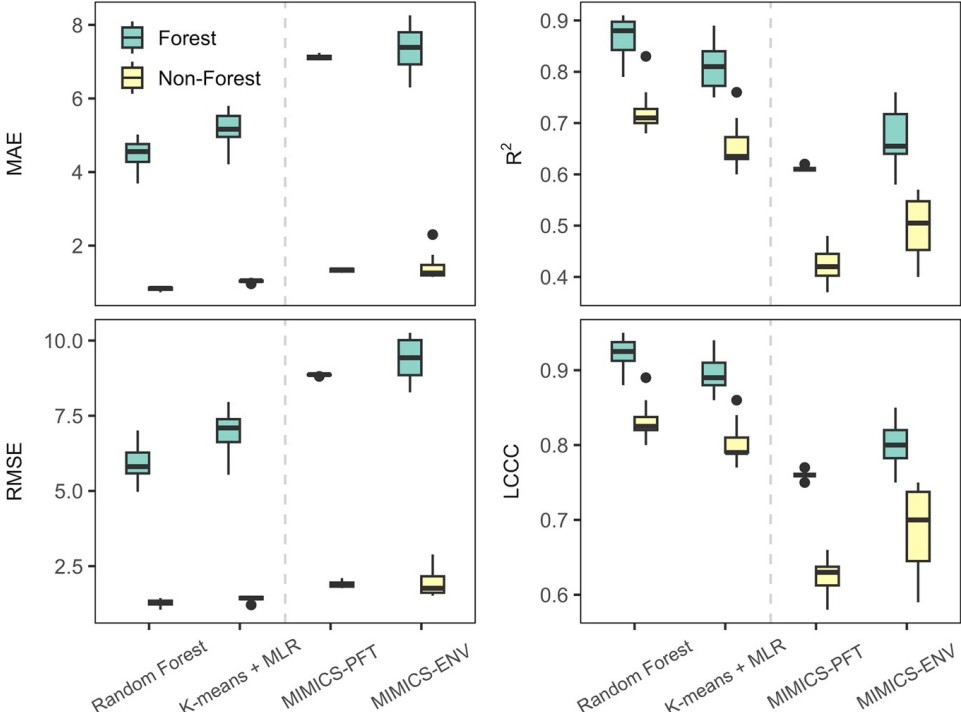

**Figure 6**. Performance metrics of SOC concentration predictions for forest and non-forest (woodland, shrubland and grassland) soils in test (out-of-sample) data. Unit for MAE and RMSE is g C/kg soil. Centre line represents median value, and upper and lower box boundaries represent third and first quartile. Whiskers extend to the smallest and largest values within 1.5 times the interquartile range.

## 3.4. Estimations of terrestrial SOC stocks

Descriptive statistics of predicted terrestrial SOC stocks at 0-30 cm soil depth are shown in Table 3. Forests have the largest mean SOC stocks ranging from 70.3 to 113.9 t/ha by all models, and shrubland is estimated to have the lowest mean SOC stocks. The distributions of predicted continental SOC stocks by all models are positively skewed with most estimated SOC stocks less than 50 t/ha (Figure 7a), and SOC stocks at peak density predicted by MIMICS-ENV and MIMICS-PFT are smaller than those predicted by machine learning approaches. The maximum value of SOC stocks predicted by all models vary considerably.

As expected, all models consistently projected greater SOC stocks in the southeast region, southwest corner and Tasmania, and consistently indicated lower SOC stocks in central and western Australia (Figure 7b). Notably, MIMICS-ENV depicted a pronounced deficit of SOC in central Australia, a distinctive pattern compared to the predictions of other models. Among the models, K-means coupled with multiple linear regression consistently provided the highest SOC estimations across all vegetation types, while MIMICS-PFT model consistently yielded the lowest mean SOC stocks.




The ensemble estimate of SOC stocks (Figure 7c) shows a similar distribution pattern as that
generated by single approach. The SOC stocks of the ensemble range from 9.3 to 180.4 t/ha
with an average value of 30.1 t/ha.  The standard deviation across the four estimates (Figure
7d) is positively correlated with the ensemble mean estimate. That is, soils with higher SOC
stocks exhibit greater variability in SOC predictions among different models. Note also that the
variability of estimates tends to be smaller in areas with denser numbers of observations.

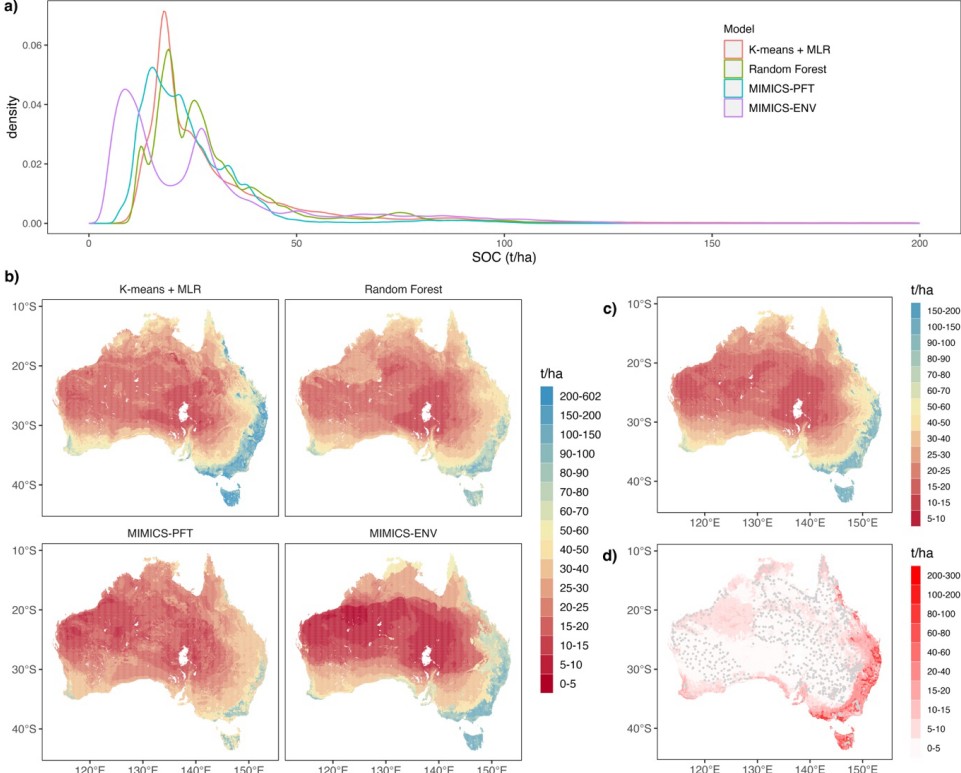

**Figure 7**. Predicted Australian terrestrial SOC stocks (t/ha) for top 30 cm soil and ensemble statistical
characteristics: a) density plot of estimated terrestrial SOC stocks by all models, noting that only stocks less
than 200 t/ha are shown for better comparison of the distribution; b) Predicted SOC stocks for each model;
c) Predicted SOC stocks of the ensemble; d) standard deviation stocks within the ensemble. Grey points
represent locations of SOC observations.





**Table 3**. Descriptive statistics of predicted terrestrial SOC stocks (t/ha) at 0-30 cm soil. Min. and Max. are
minimum and maximum value, respectively. 1st Qu and 3rd Qu are first and third quartile, respectively.

| | PFT | Min. | 1st Qu | median | mean | 3rd Qu | Max. |
|---|---|---|---|---|---|---|---|
| K-means + MLR | grassland | 4.2 | 17.9 | 21.2 | 41.5 | 42.5 | 601.1 |
| | shrubland | 7.2 | 16.4 | 19.3 | 23.6 | 24.4 | 472.2 |
| | woodland | 7.1 | 20.1 | 26.1 | 33.3 | 33.7 | 483.1 |
| | forest | 18.0 | 51.3 | 95.2 | 113.9 | 153.4 | 474.0 |
| | all | 4.2 | 18.1 | 23.6 | 38.2 | 16.7 | 601.1 |
| Random Forest | grassland | 10.4 | 18.5 | 26.0 | 30.4 | 37.2 | 125.3 |
| | shrubland | 10.3 | 17.0 | 19.6 | 21.4 | 24.4 | 104.4 |
| | woodland | 10.5 | 20.3 | 25.8 | 28.2 | 32.4 | 122.1 |
| | forest | 29.3 | 55.0 | 82.3 | 78.4 | 97.0 | 161.7 |
| | all | 10.3 | 18.9 | 25.0 | 29.8 | 33.7 | 161.7 |
| MIMICS-PFT | grassland | 10.8 | 16.4 | 24.1 | 25.1 | 33.3 | 58.7 |
| | shrubland | 6.5 | 12.2 | 15.5 | 16.5 | 20.6 | 56.5 |
| | woodland | 7.8 | 17.4 | 21.2 | 22.1 | 25.9 | 61.4 |
| | forest | 17.9 | 44.5 | 77.4 | 70.3 | 88.5 | 109.9 |
| | all | 6.5 | 15.7 | 21.2 | 24.3 | 28.9 | 109.9 |
| MIMICS-ENV | grassland | 4.2 | 7.9 | 15.4 | 26.9 | 37.6 | 124.0 |
| | shrubland | 4.4 | 9.9 | 13.4 | 18.6 | 25.3 | 131.9 |
| | woodland | 4.9 | 14.7 | 26.3 | 28.4 | 31.3 | 131.6 |
| | forest | 9.6 | 53.1 | 92.4 | 81.6 | 106.5 | 134.1 |
| | all | 4.2 | 10.5 | 21.9 | 28.1 | 33.2 | 134.1 |


## 4. Discussion

### 4.1. Relative importance of predictors on SOC variations


Extensive research has been conducted to discern the factors that govern SOC content/stocks.
Among the commonly employed predictors for SOC variations, climate, organisms,
topography, parent material, and soil properties are prominent (Wiesmeier et al., 2019). Within
this study, we conducted a comparative assessment of the significance of key variables, namely
MAT, MAP, NPP, soil clay content and bulk density, in driving variations in SOC in Australia.
While the number of predictors utilized in our approach is fewer than that employed in most
digital mapping methodologies, its strength lies in the potential for a more direct comparison
between empirical and process-based models.

Soil bulk density is the most important driver of SOC concentration at the continental scale
(Figure 3). The role of soil bulk density in SOC has been noted before in eastern Australia
(Hobley et al., 2015). Soil bulk density is mainly a function of the parent material, soil genesis
as well as soil aggregate formation (Don et al., 2007). A soil with reduced density exhibits
superior structural organization and an expanded surface area, facilitating enhanced retention
of organic carbon (Lobsey and Viscarra Rossel, 2016). Subsequently, with a slightly lower value
of importance than soil bulk density, MAT emerges as the second most influential factor
governing SOC variations, followed by NPP, MAP, and clay content. This sequence of
significance diverges from the findings of Walden et al. (2023), where the order of importance
was observed as NPP > clay content > MAP > MAT on a continental scale in Australia. The





number of predictors used in their study is much more than that in our study, which may affect the contribution of given predictors in SOC variation (Guo et al., 2019). This discrepancy might however be attributable to the utilization of observations encompassing both terrestrial and blue carbon ecosystems in their study. Clay emerges as key driver mainly in the groups where aquatic plants (e.g., seagrass, tidal marsh) appeared. The more extensive dataset encompassing the eastern coastline, characterized by greater variability and abundance of NPP input, potentially elevates NPP to a dominant role in influencing SOC variations within their study.

For SOC in different vegetation types (Figure 3), soil bulk density and MAT are more important than other factors in forest, and all factors except clay content showed similar importance in driving SOC in grassland. NPP and MAP dominate the SOC variations in woodland and shrubland. Climate conditions exert their impact on SOC in all vegetation types. It was proposed that the primary climatic determinant of SOC variation hinges on the primary constraint affecting SOC production and turnover (Hobley et al., 2016). In this study, most shrublands and woodlands are distributed in arid and semi-arid regions characterized by limited precipitation, which leads to water stress in surface soil, limiting plant productivity and reducing soil C input (Hobley et al., 2015). Consequently, MAP and NPP exhibited relatively higher influence on SOC variations. In contrast, forest SOC observations are mainly distributed in areas with relatively lower temperatures, therefore experience constrained microbial metabolism, leading to reduced decomposition rates and the high accumulation of SOC (Wynn et al., 2006). Consequently, MAT emerges as a key factor influencing SOC variations in forests. Furthermore, it is noteworthy that soil bulk density plays a crucial role in determining SOC distribution within forest ecosystems, where it is significantly lower compared to other vegetation types. This lower soil bulk density likely facilitates the formation of microaggregates and enhances the preservation of SOC within the soil matrix (Bronick and Lal, 2005). Consequently, it effectively contributes to elevated SOC concentration levels in forested areas.

PFTs are the only categorical predictor for SOC concentration in this study. SOC is mainly derived from plant C input through above-/belowground tissues, and SOC turnover and storage are influenced by plant traits like plant growth rate and chemical and physical composition (De Deyn et al., 2008; Faucon et al., 2017). With shared representation of similar plant traits, PFTs are widely used in process-based models (Poulter et al., 2015; Famiglietti et al., 2023). It was found that the vertical distribution of SOC is highly related to PFTs due to the different root distribution and above- and belowground allocation (Jobbágy and Jackson, 2000). However, our study is limited by the absence of SOC observations at multiple soil depths, restricting the analysis to the spatial distribution of SOC at 30 cm soil depth. The influence of PFTs on SOC concentration at this particular depth appears relatively insignificant (Figure 3), casting doubt on the effectiveness of optimizing parameters of process-based model for individual PFTs (Cranko Page et al., 2023). Considering this, employing the top 3 influential abiotic predictors, soil bulk density, MAT, and MAP, we partitioned all observations into six distinct clusters using K-means. It was anticipated that SOC ranges within each cluster would be narrow due to the high similarity of these three predictors within each group. However, the distribution of SOC




in clusters 2, 4, and 6 exhibited considerable variability (Figure 4). Given that these clusters are predominantly composed of forest ecosystems, it becomes apparent that these three abiotic factors alone are insufficient to fully characterize the intricacies of forest SOC concentration. It was found that elevation and evapotranspiration also drive the variation of forest SOC in Australia (Walden et al., 2023), and taking them into account might potentially increase the predictability of forest SOC.

## 4.2.    Model evaluation and comparison

Although the predictors used for machine learning models are not exactly same as the inputs of MIMICS, the missing factors (e.g., MAP) were used for parameter optimization of MIMCS-ENV, making the predictions dependent on similar information and so comparable to some extent.  Besides, our study presented clear evaluation metrics for out-of-sample validation, enabling a more robust assessment of model performance when applied to new datasets.

Based on the performance metrics of test data, the machine learning models performed remarkably well (Figure 5). The $R^2$ suggest that both two machine learning models can explain more than 90% of SOC variability across sites, and random forest did the best job with greatest $R^2$ and LCCC, and lowest MAE and RMSE. Random forest algorithms were widely adopted in predicting spatial-temporal SOC dynamics and produced moderate well performance regionally and globally. For example, Wang et al. (2022) applied random forest to estimate SOC stocks in south-eastern Australia and explained 69% of the variation of current SOC stocks. Nyaupane et al. (2023) trained a random forest model using global SOC observations and explained 61% of SOC variation. The good performance of random forest might be attributed to reduced susceptibility to over-fitting and better capacity to manage the hierarchical non-linear relationships that exit between SOC and environmental predictors (Wang et al., 2018b). Other machine learning methods have been applied to predict continental SOC stocks in Australia. For example, Walden et al. (2023) trained regression-tree algorithm CUBIST to predict SOC stocks at 30 cm soil using the harmonised datasets. The mean LCCC and RMSE for out-of-sample validation in their study was 0.78 and 0.20 respectively when $\log_{10}$ transformed SOC (t/ha) values used. Wadoux et al. (2023) applied quantile regression forest to predict SOC stocks at multiple soil depths. The prediction accuracy decreased dramatically for deeper depth intervals with the greatest $R^2$ (0.53) at 0-5 cm soil. The better results in this study may be attributed to the removal of cropland ecosystems, which are clearly highly managed and so less predictable. Agricultural practices greatly affect SOC content in Australia and add the complexity to the relationship between SOC and environmental factors (Luo et al., 2010). Models using environmental predictors without representation of land use management are unlikely to be able to fully capture the SOC dynamics in croplands (Abramoff et al., 2022).

Although MIMICS made less accurate estimations of SOC than machine learning models, it did well at continental scale with mean $R^2$ at 0.82 and 0.84 for MIMICS-PFT and MIMICS-ENV, respectively (Figure 5). Georgiou et al. (2021) found that there was a mismatch between



observations and MIMICS in the role of environmental controls in explaining SOC variability
at global scale. NPP and MAT had the most explanatory power for SOC stocks from MIMICS,
while clay content had the most explanatory power for SOC observations, which limits the
predictability of SOC using MIMICS in their study. However, in our study, NPP and MAT rather
than clay content played a greater role in observed SOC variations, perhaps contributing to a
better performance of MIMICS in Australia. The modest performance of MIMICS relative to
machine learning models could potentially be attributed to the absence of explicit representation
of MAP. The augmentation of MAP within parameter optimization in MIMICS-ENV did allow
improved performance compared to MIMICS-PFT, particularly within non-forest regions
where the importance of MAP rivals or surpasses that of temperature. Precipitation is a
determinant of plant productivity, especially in arid and semi-arid regions. Besides, arid regions
with limited precipitation are characterized by lower weathering rate limiting the formation of
mineral-associated soil carbon (Doetterl et al., 2015). Hence, we assume that introducing the
effect of moisture to MIMICS could contribute to more accurate prediction of SOC, as
compared with just taking MAP into account for parametrization, especially in arid and
semiarid regions.
All models produced lower MAE and RMSE for non-forest SOC but higher $R^2$ and LCCC for
forest SOC (Figure 6). SOC in forest is more abundant and variable compared to SOC in other
vegetation types even when climate conditions are similar, which constrains the accuracy of
forest SOC estimation. However, the higher $R^2$ and LCCC mean that all models show higher
ability to predict forest SOC using environmental predictors. Forests, given they are less
perturbed ecosystems, might afford greater SOC predictability due to the reduced influence of
direct anthropogenic disturbances. This stands in contrast to ecosystems like grasslands,
shrublands, and woodlands, predominantly situated in Australian rangelands where extensive
grazing constitutes the predominant agricultural practice. Primarily, grazing leads to a reduction
in soil carbon influx originating from aboveground biomass. Moreover, the cascading effects
of grazing extend to potential alterations in plant composition and structural attributes, inducing
consequential shifts in litter properties that modulate soil carbon decomposition kinetics (Lunt
et al., 2007; Bai and Cotrufo, 2022). This intricate interplay of grazing-induced disturbances
introduces a layer of complexity to SOC predictions. The disturbances triggered by grazing
manifest in soil carbon pools, leading to a state of disequilibrium rather than adhering to the
assumption of SOC convergence toward equilibrium, as embraced in this study's framework.
Notably, forests, as relatively undisturbed natural ecosystems, demonstrate a better coherence
with the equilibrium assumption, rendering their SOC more amenable to prediction through
environmental drivers.
4.3.    Spatial prediction of SOC stocks
We produced gridded SOC stocks across Australia using the models validated in this study and
an ensemble estimate as the average of four models (Figure 7). Among the models, K-means
coupled with multiple linear regression produced the largest mean SOC stocks both at



continental scale and for all vegetation types. In contrast, random forest and MIMICS with
different parameterization approaches produced more conservative SOC stock estimations. The
mean terrestrial SOC stocks estimated by random forest and MIMICS are comparable with that
estimated by Australian baseline map, which was generated using machine learning algorithm,
reporting the mean SOC stocks at 29.7 t/ha with 95% confidence limits of 22.6 and 37.9 t/ha
(Viscarra Rossel et al., 2014). However, SOC stocks might be underestimated by these methods
because of the scarcity of data from the most productive temperate forest both in the baseline
map (Bennett et al., 2020) and in our study. Parameter optimization process of MIMICS and
the training process of random forest are greatly affected by data used to train the model. Most
SOC observations in this study were sourced from arid and semiarid regions, characterized by
limited SOC content. As a result, the models' ability to predict SOC stocks beyond the observed
data range is somewhat constrained. PFTs was found to be less important than other
environmental factors in driving spatial SOC variations (Figure 3), so it was perhaps not
surprising that applying parameters optimized for each plant functional type to the regions with
same PFT but broader climate conditions led to inferior results than applying parameters
optimized for each environmental group.
The utilization of linear regression in K-means + MLR generated SOC estimates beyond the
range of observations, particularly in eastern Australia where environmental conditions deviate
from the training data. The mean SOC stocks estimated by K-means + MLR (38.2 t/ha) are
higher than those of the other models employed in this study, and align closely with the mean
value 36.2 t/ha reported by Walden et al. (2023) who updated the Australian baseline SOC map
(Viscarra Rossel et al., 2014) by incorporating additional SOC observations from forests and
coastal marine ecosystems. However, caution is required when interpreting extreme values
derived from the K-means + MLR, such as the instance of grassland SOC stocks reaching 601
t/ha (Table 3). These values raise concerns about the reliability of this approach when
extrapolating out-of-sample. Though there is a positive relationship between NPP and SOC
observations in this study, SOC accumulation cannot continuously increase linearly in the
regions where environmental conditions seem highly conducive to SOC formation. The greater
amount of carbon input in eastern Australia might trigger the acceleration of microbial
decomposition because of a priming effect, and lead to a decreased accumulation of SOC stocks
(Ren et al., 2022). The existence of SOC saturation also implies that SOC cannot be
accumulated without limit (Georgiou et al., 2022; Viscarra Rossel et al., 2023). In light of these
complexities, applying linear regression to predict SOC content, especially under the extreme
environmental conditions, should be undertaken with care.
Continentally, higher SOC was estimated for the southwest corner and southeast Australia
(Figure 7), aligning with other SOC maps for Australia (Wadoux et al., 2023; Walden et al.,
2023). These regions are characterized by lower temperature and higher precipitation, therefore
high SOC accumulation appeared because of stimulation of NPP by moisture and the
constrained microbial metabolism in low temperatures. Forest has the largest mean SOC stocks
ranging from 70.3 to 113.9 t/ha estimated by four models in this study. Around 75% of the forest



SOC is from soil under Eucalypt open forest, and mean SOC stocks under this type of forest
were estimated to be 87.5 t/ha (63.8 -119.6 t/ha for 95% confidence interval) (Walden et al.,
2023). Shrublands are estimated to have the lowest mean SOC stocks, and more than 90% of
shrub SOC observations are from soil under Acacia shrubland and Chenopod shrubland, which
rank at the bottom of SOC stocks among different vegetation types (Walden et al., 2023). The
low SOC in shrubland is probably due to low carbon input because of limited rainfall (MAP <
280 mm). Though the mean SOC stocks in non-forest regions are much smaller than that for
forest, the greater area of vegetation cover results in considerable total SOC stocks, highlighting
the importance of carbon building and maintaining via improved managements in these areas.
Greater variability of SOC estimates among different models appears in the regions where SOC
stocks are higher (Figure 7). The sparsity of SOC observations is a primary contributor to the
uncertainties associated with SOC estimates in these regions, highlighting the importance on
continual collection of data to better constrain models' behaviour. This imperative is especially
pronounced in regions covered by forests, as forested soils exhibit substantial SOC stocks,
amplifying the significance of abundant and accurate data acquisition in these specific
ecosystems.
## 5. Conclusion
We compared the performance of two machine learning models, and one process-based
microbial model employing distinct parameterization approaches, to explore the diversity of
SOC estimates within a 30 cm soil depth across Australia. Results highlight soil bulk density
and MAT as predominant factors governing SOC concentration variations, both on a continental
scale and within forest and grassland ecosystems. Conversely, NPP and MAP exhibit greater
significance in driving SOC variations within shrubland and woodland soil. Our study
underscores the importance of including the influence of appropriate environmental factors in
process-based models in different environments.
Validation results affirm that with appropriate filtering of data (e.g. removing highly managed
crop ecosystems) models can predict SOC at a continental scale with reasonably high reliability,
achieving explained variances exceeding 80% for out-of-sample test data, with random forest
showing highest prediction accuracy. Notably, all models show higher $R^2$ in prediction of SOC
under forest than under non-forest vegetations. MIMICS, with parameters optimized for
different environmental clusters, performed better in SOC prediction than MIMICS with
parameters optimized for different PFTs, especially in non-forest regions.
All models broadly agree on the spatial distribution of SOC, with higher SOC stocks
concentrated in the southeast and southwest regions of Australia. However, the variations in
estimated values need to be acknowledged, particularly in highly productive regions. Among
these estimates, K-means algorithm coupled with multiple linear regression yields the highest
mean SOC stocks estimate, while the MIMICS-PFT model generates the most conservative
estimate. Considerable disagreement of the maximum and minimum SOC stock values



predicted by all models exists partly because models are less constrained by observations in
these environments, highlighting the need for continued observational campaigns.
Our investigation has revealed significant disparities in estimated SOC stocks when different
methodologies were employed. This highlights the need for a critical re-evaluation of land
management strategies that heavily depend on SOC estimates derived from a single approach.
The incorporation of an ensemble of SOC estimates is more likely to effectively capture
elements of the uncertainty associated with SOC estimations, providing a more robust basis for
informing strategies in soil carbon management and climate change mitigation.

## Code availability

Source Code of vertically resolved MIMICS can be accessed at the CSIRO data portal
https://doi.org/10.25919/843a-w584 (Wang et al., 2021). Codes for data analysis and machine
learning can be accessed by contacting the correspondence author.

## Data availability

The SOC observations from VR dataset are not publicly available but are available from
Raphael A. Viscarra Rossel (r.viscarra-rossel@curtin.edu.au) on reasonable request. All other
data used in this study are publicly accessible and the specific references of these databases are
provided in Section 2.4.

## Author contribution

Conceptualization: LW, GA, Y-PW, AP; Methodology: LW, GA, Y-PW; Investigation: LW,
RAVR; Formal analysis and Visualization: LW; Writing-original draft preparation: LW;
Writing-review & editing: LW, GA, Y-PW, AP, RAVR.

## Competing interests

The co-author Raphael A. Viscarra Rossel is a member of the editorial board of SOIL.

## Acknowledgements

LW thanks the China Scholarship Council and the University of New South Wales for financial
support during her PhD study. RAVR and Y-PW thank the Australian Research Council's
Discovery Projects scheme (project DP210100420) for funding. LW, GA and AP thank the ARC
Centre of Excellence for Climate Extremes for supporting this work (CE170100023).



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
