# Peer review of "An ensemble estimate of Australian soil organic carbon using machine learning and process-based modelling"

_EGUsphere, 2023_

## Author Comment (AC4)

**The authors applied different models to predict the spatial variability of SOC concentration in different Australian ecosystems, using 1000 in-situ observations. The article is well written and structured and I think it should be considered for publication at EGUsphere. However, there are some flaws that should be addressed by the authors before publication.**

We would like to thank the reviewers for their time and feedback on this manuscript. Please find our point-to-point responses below.

1. **First, the authors use SOC stocks to calculate SOC concentration using bulk density from the Soil and Landscape Grid National Soil Attributes Maps (SLGA). This is counter intuitive since SOC stocks cannot be measured but are usually estimated from SOC concentration and bulk density measurements. Why the authors don't use source data for SOC concentration? Was the SOC stock from their database initially calculated using the same bulk density data that they use?**

It's correct that we used bulk density from the SLGA data, however, our explanation wasn't clear, and we apologise for that. Let us explain. We did not use direct measurements of bulk density because they were not available. The recalculation of SOC concentration from SOC stocks and bulk density was to better compare the results from the simulations. We simulated SOC flux with MIMICS in mg/cm$^3$, and one of our datasets only had SOC stocks in t/ha. Unfortunately, the lack of new soil measurements over the country means that we can really only use datasets that are currently available. To compare the simulated SOC to the observations, we converted the units of both simulated SOC and observed SOC to g C/kg soil using the SLGA-derived bulk densities at each site. These unit conversions will not affect the results of MIMICS.

To compare the performance of MIMICS and machine learning models using metrics such as RMSE and MAE which are unit-specific, we also used SOC in g C/kg soil to train the machine learning models. We will clarify this in the revised manuscript.

2. **One of the main findings is that bulk density is one of the main predictors of SOC concentration. However, the common understanding is that SOC concentration is a predictor of bulk density so I think they should refine their modelling scheme to avoid biases in their result and revise the discussion on this part.**

Yes, that is correct. There is a clear (and known) negative relationship between SOC and bulk density. For example, higher bulk density usually corresponds to less porosity restricting plant root growth, oxygen availability to soil microbial communities and then SOC dynamics. Conversely, soil that is less dense has greater porosity, allows water infiltration, oxygen diffusion, root growth, microaggregate formation and enhances SOC dynamics. Therefore, it is reasonable to use bulk density as a predictor of SOC in our machine learning models. We'll revise the discussion to explicitly consider the relationship.

3. **Second, they use a vertically discretized version of MIMICS, but then only show results for the top 30 cm. What was the rationale behind using this vertically discretized model? I think it would be interesting to show the results – at least in the supplementary material – on the SOC concentration also for deeper layers. Even if they won't be able to compare it with observations, if possible, they can discuss their findings in comparison to the literature.**

Thanks for the suggestion. We used the vertically discretised version of MIMICS in this study mainly to consider the carbon loss from top 30 cm soil to deeper layers due to vertical transport

processes. This aspect is lacking in the default version of MIMICS, where the only output of SOC is microbial respiration.

We compared the statistics between simulated SOC concentration in 30-150 cm soil by MIMICS (calculated as the mean value of 30-150 cm at 10 cm thickness) and SOC concentration extracted from a digital soil mapping product SLGA (calculated as weighted average of 30-60 cm, 60-100 cm and 100-200 cm soil). The mean value of SOC simulated by MIMICS-PFT (5.39 g C/kg soil) and MIMICS-ENV (6.54 g C/kg soil) are higher than that of SLGA (2.65 g C/kg soil) but lower than the mean value of the map representing 95% confidence interval of SLGA (6.99 g C/kg soil).

As noted by the reviewer, while we applied the vertically discretised version in this study, we only had observations for top 30 cm soil. The deeper layers were actually excluded from our parameter optimization process, they were not included in the calculation of the cost function. As SOC content in 0-30 cm is usually greater than that in deep soil layers, applying parameters optimized for 0-30 cm layer to deeper layers is likely to overestimate SOC at depth, making the predictions highly uncertain and unreliable. Therefore, we think it's less meaningful to present the results for deep layers in this paper. If the editor believes this to be important, we could compare statistics (e.g., min, max, mean value) of the MIMICS predictions of SOC in deeper layers to those extracted from the SLGA in a supplementary information file.

**General comments**

4. **Throughout the text, it is often unclear whether they refer to SOC stocks or SOC concentration. Please add this information each time when this is appropriate.**

Thanks, we'll clarify it in the revised manuscript.

5. **In the discussion, they should make an effort to better compare their results to the existing literature.**

Thanks, this was noted by Reviewer 1 as well, and as noted in our response there we'll improve our comparison with existing results and extend the discussion to include this.

6. **They chose to call one of their clustering groups plant functional types, but what they use are actually ecosystems. Please change this name throughout the text.**

Thanks for your suggestion regarding the classification of observations into ecosystems rather than plant functional types. However, we suggest that in this context "plant functional types" are appropriate, for two reasons,

1) The vegetation type is extracted from the National Vegetation Information Systems (NVIS) where the vegetation is divided into 32 "Major Vegetation Groups'. We aggregate them into four main types to guarantee sufficient observations for each group for model optimization. The aggregation is based solely on vegetation, without the consideration about other environmental characteristics.
2) "Plant functional types" are the classifications used in almost all process-based land models (e.g., land surface models) to represent plant and ecosystem diversity, so that using plant functional types here aligns better with existing literature and modelling approaches. This includes some existing applications of the MIMICS model (e.g. https://doi. org/10.1029/2020JG006205)

7. **The authors should add a map of the sites used and specify how many sites did they have for each ecosystem.**

Thanks for the suggestion. Figure 2 shows the spatial distribution and the number of sites in each plant functional type. We'll add a similar map for each environmental group in the revised manuscript.

**Specific comments**

8. **L23 I think plant functional types is not really a good definition of what you used. You used ecosystems rather than PFT.**

This is the same comment as 6 above. Please see our reply above.

9. **L28 when you say non-forest soils, you need to specify that you're not including croplands.**

Thanks, we'll clarify that in the manuscript revision.

10. **L45-46 reference needed.**

Thanks, we'll add references here (Jackson et al., 2017, https://doi.org/10.1146/annurev-ecolsys-112414-054234; Stockmann et al., 2013, https://doi.org/10.1016/j.agee.2012.10.001).

11. **L90-91 but also the parametrization and the lack of model validation with data (see Le Noe et al, 2023)**

**Le Noë, Julia, Stefano Manzoni, Rose Abramoff, Tobias Bölscher, Elisa Bruni, Rémi Cardinael, Philippe Ciais, et al. "Soil Organic Carbon Models Need Independent Time-Series Validation for Reliable Prediction." Communications Earth & Environment 4, no. 1 (May 8, 2023): 158. https://doi.org/10.1038/s43247-023-00830-5.**

Thanks for the addition. We'll revise it.

12. **L93 change "models has rarely" with "models have rarely"**

Thanks, we'll amend it.

13. **L91-94 Please add: and the difficulty of constraining the parameters of microbial explicit C models with relevant data on microbial properties, for example extracellular enzyme activities which are very hard to measure but also very important for these type of models.**

Thanks, we'll add it.

14. **L95 change "understanding on the" with "understanding of the"**

Thanks, we'll revise it.

15. **L96-98 You can also cite: Todd-Brown, K. E. O., J. T. Randerson, W. M. Post, F. M. Hoffman, C. Tarnocai, E. A. G. Schuur, and S. D. Allison. "Causes of Variation in Soil Carbon Simulations from CMIP5 Earth System Models and Comparison with**

Observations." *Biogeosciences* 10, no. 3 (March 13, 2013): 1717–36. https://doi.org/10.5194/bg-10-1717-2013.

Thanks, it's really a good paper and we'll cite it here.

**16. L107-108 please specify what you mean with moderately well.**

Thanks, we'll revise the sentence to make it clear.

**17. L122 add "low" before amount of data.**

Thanks, we'll add it this.

**18. L129 add in forests, woodlands, shrublands and grasslands.**

Thanks, we'll add that.

**19. L143 change The" MIcrobial-MIneral Carbon Stabilization (MIMICS)" with "MIMICS"**

Thanks, we'll revise it.

**20. L144 change "soil carbon" with "SOC".**

Thanks, we'll revise it.

**21. L158 change "soil carbon" with "SOC".**

Thanks, we'll revise it.

**22. L180 average conditions during what period?**

We used climate data from 1991 to 2020 in this study (Table 2).

**23. L181 please specify what you mean by "a sensible range".**

It means "a reasonable range" – we referred the papers but it was based more on experience to set the values because the pools are conceptual and cannot perfectly correspond to observed SOC fractions. We'll be more explicit about our choices here in the revised manuscript.

**24. L194-196 "We trained the RF model with different numbers of trees": how many?**

We set the number of trees as 100, 200, 300, 400 and 500, and the performance of random forest didn't vary much with different number of trees. We therefore chose a tree number of 200 in this study. We'll note this in the revised manuscript.

**25. L203 use aggregate instead of segregate?**

Thanks, we'll modify this.

**26. L221 what predictors? please specify.**

Thanks, we'll specify predictors here.

**27. L223 Change permutation variable importance with PVI.**

Thanks, we'll revise it.

**28. L226 Change random forest with RF.**

Thanks, we'll revise it.

**29. L246 in Table 1, add the units of the parameters.**

The parameters are scaling factors and are therefore unitless.

**30. L250 change "The second approach was taking" with "The second approach consisted in taking".**

Thanks, we'll amend this to 'the second approach used the most…'.

**31. L251 specify which were the main predictors.**

Thanks, we'll specify that.

**32. L269-270 reference needed for SILO database.**

We show the reference in Table 2. For better readability we'll move it to main text.

**33. L275-278 In grasslands, or even in managed forests, NPP should be corrected by removing the fraction of NPP that is appropriated by human activities. Also, there might be high variabilities due to C input sources. The best would be to use different sources to see the uncertainties, but at least this should be discussed.**

Thanks for the good suggestion here. We agree that the sources of NPP data will cause uncertainties in SOC estimations, and we'll add this part to discussion.

**34. L281 reference needed for NVIS.**

Sorry for the missing reference. We'll add it.

**35. L285 reference needed for SLGA.**

We show the reference in Table 2. For better readability we'll move it to main text.

**36. L309-311 do these datasets use comparable methodologies for soil analysis?**

The methods used to measure SOC concentration in the laboratory are different because they are from different projects, see Viscarra Rossel et al., (2019) and Bissett et al., (2016). However, the methods used to process and harmonise the datasets are the same and follow the approach in Viscarra Rossel et al., (2014). Briefly, these two datasets used are harmonised to represent SOC in

the 0-30 cm layer using natural cubic splines for profiles with observation at more than 3 layers and a log-log model to profiles with observation at two layers. We will clarify that this is what we did.

**37. L312 change "algorithm" with "equation".**

Thanks, we'll revise it.

**38. L317-318 specify the values used for S and I.**

The equation is applied to each site, so the parameters S and I are site-specific.

**39. L 332-336 move (a) to the beginning of the sentence. same with (b). Also, you need to add (a) and (b) in the figure.**

Thanks, we'll revise it.

**40. L340-342 Change the sentence as follows: For machine learning models, all observations were randomly separated into a training and test datasets, with […].**

Thanks, we'll revise it.

**41. L343 what groups?**

Each plant functional type or environmental group divided using K-means. We'll clarify this in the revised manuscript.

**42. L340-345 And then what values did you keep after cross-validation?**

We selected the set of parameters which made MIMICS perform best on out-of-sample test data based on the metrics, please see L361-364 for details. And we'll add the optimized value of each parameter used to predict SOC at the continental scale in a new table.

**43. L370 change "Permutation Variable Importance (PVI)" with PVI.**

Thanks, we'll revise it.

**44. L411 change random forest with RF.**

Thanks, we'll revise it.

**45. L415 add "but with a higher variability across sites" at the end of the sentence.**

Thanks, we'll add it.

**46. L455 change "approach" with "model".**

Thanks, we'll revise it.

**47. L457-459 but what about the relative standard deviation. is it still positively correlated with the mean? it's logic that higher mean ==> higher standard deviation in absolute terms, but it would be interesting to see if also the relative standard deviation is higher.**

Thanks, it's really a good suggestion, we'll add the map of relative standard deviation.

**48. L475 In table 2, add the results for the multi-model mean.**

Thanks, we'll add that.

**49. L483-484 add SOC spatial variations?**

Thanks, we'll revise it.

**50. L488 what does it entail in terms of uncertainty the fact that the number of predictors you used was lower than that employed in most digital mapping?**

We think the uncertainty caused by the limited number of predictors is relatively small.

Most of studies that used machine learning for digital mapping of SOC typically employed different categories of predictors, including climate, organisms (e.g., plants), soil properties, terrain attributes (e.g., elevation) and land-use. In each category they usually selected several proxies, for example, temperature, precipitation, and evapotranspiration were widely used to represent climate conditions. Usually more than 20 predictors were employed in the digital mapping studies. However, some predictors contributed similarly to the predictability of SOC because of their collinearity, and some predictors contributed little because they were less important. Therefore, removing the redundant predictors and keeping only the most important ones (as assessed, for example, by random forest variable importance indicators) may actually provide not just comparable skill, but increased interpretability of SOC variations.

In our study, we used mean annual temperature, mean annual precipitation, soil clay content, bulk density, and NPP to train the machine learning model. These predictors were widely found to be important in SOC variations. Taking them as predictors, our random forest model showed good performance in predicting SOC concentration with $R^2$ greater than 0.9 for out-of-sample test data. In this case we think the predictors we used are sufficient to interpret SOC variations in Australia. While including additional predictors clearly may improve machine learning performance, the enhancement will not be large, given that $R^2$ is already greater than 0.9.

Equally important is limiting predictors to be comparable to the predictors given to the MIMICS model. This allows us to understand how well MIMICS is utilising the information available to it. Therefore, we opt to retain a limited number of predictors to facilitate a more meaningful comparison among different models.

**51. L491 is there a bias because you calculated concentration from SOC stocks and bulk density?**

We used SOC concentration (g C/kg soil) to train random forest model in order to compare models' performance based on metrics like RMSE and MAE (please see our reply to the first comment above). To test if the importance of bulk density on SOC concentration is universal, we

1) estimated variable importance using only forest SOC observations from BASE dataset where SOC is originally reported in concentration (%). Bulk density also ranked first among all predictors.
2) Estimated variable importance using a global SOC concentration dataset (WOSIS, https://www.isric.org/explore/wosis), and bulk density also ranked first in SOC concentration (unpublished result).
3) Referred to papers (e.g., https://doi.org/10.1007/s11104-015-2380-1) that also found the importance of bulk density on SOC concentration in Australia.

Therefore, we think the bias due to the calculation of SOC concentration using bulk density is small.

**52. L493-494 And land-use?**

Yes land-use significantly affects soil bulk density. We'll mention it here.

**53. L501 change much more with much higher.**

Thanks, we'll revise it.

**54. L500-502 what other attributes did they use? does it mean that the number of predictors you used is not sufficient? Can you test that?**

They used around 30 predictors relating to climate, soil properties and so on. The predictors we used are sufficient in this study, please see our reply to (50) above for details.

**55. L517-518 add "in these ecosystems" at the end of the sentence.**

Thanks, we'll revise it.

**56. L521-523 due to what?**

Soil bulk density in forest is much lower than that in soils under other vegetation types. This is a finding based on the analysis of observations. It may be attributed to the root penetration, the formation of aggregate and stable structure due to higher content of soil organic matters in forests. And organic matters have much lower bulk density than other soil constitutes such as clay, silt and sand, therefore, forest soils with high SOC content usually have lower bulk density than non-forest soils. On the other hand, lower bulk density improves the oxygen availability to microbial communities, accelerating their activities for carbon stabilization. We'll make the relationship between SOC and bulk density explicit in the revised manuscript.

**57. L538 change process-based model with process-based models.**

Thanks, we'll revise it.

**58. L558 remove "two".**

Thanks, we'll remove it.

**59. L562 change moderately well with moderately good.**

Thanks, we'll revise it.

**60. L568 change "exit" with "exist".**

Sorry for the typo here. We'll amend it.

**61. L571 change "at 30 cm soil" with "at 30 cm depth".**

Thanks, we'll revise it.

**62. L573 add "were" between "values" and "used".**

Thanks, we'll revise it.

**63. L582-584 where does MIMICS get wrong?**

Based on the findings in this study, we can first contribute the lower accuracy of MIMICS to the lack of explicit representation of a soil moisture effect. MAP is important for SOC variations especially in shrublands and woodlands (Figure 3) and taking it into account for parameter optimization (MIMICS-ENV) benefits the model performance (greater $R^2$ and LCCC in Figure 6) in these vegetation types. Though MAP clearly isn't the same as soil moisture, it highlights the importance of water in SOC dynamics (see L590-600 for details).

MIMICS is established based on our understanding on SOC turnover processes. However, some processes are simplified or ignored because of the lack of knowledge and observational data, as well as the increased uncertainties when new parameters are introduced. Machine learning models learn patterns from the data without relying on the explicit mechanisms of SOC dynamics, therefore can better capture and handle the complex relationships between predictors and SOC.

**64. L586-588 change the sentence as follows: In their study, NPP and MAT had the highest explanatory power for SOC stocks according to MIMICS, while clay content had the highest explanatory power according to the SOC stock(???) observations, which limits the predictability of SOC using MIMICS.**

Thanks, we'll revise the sentence here to make it clearer.

SOC is originally reported in concentration (%), but model output is in mg/cm$^3$. To compare model outputs and observations, we need to unify the unit using bulk density. When we say SOC stock observations, we mean converted observed SOC concentration to SOC stocks using bulk density to compare with model outputs. We'll make this clear in revised manuscript.

**65. L588-590 Do you mean according to the random forest model applied to predict SOC concentration?**

Yes, according to the PVI value showing the importance of predictors (Figure 3). PVI values are estimated using random forest.

**66. L602-603 which means that...**

SOC concentration in forest soils is much higher and more variable than that in soils under other vegetation types, which will likely lead to a higher MAE and RMSE. However, based on the higher $R^2$ and LCCC metrics for forest, we conclude that SOC in forest soil is more predictable because it aligns well with the observed patterns. We'll revise the discussion here to clarify this point.

**67. L603-604 what do you mean? Do you mean it limits the accuracy?**

The sentence here aims to explain why the MAE and RMSE for forest SOC is higher than those for other vegetation types (please see our reply to (66) above for details). Predictions of SOC in forest likely have larger absolute errors but also larger consistency to observed patterns. We'll rewrite this in the revised manuscript to make it clearer.

**68. L606 add "that" after "Forests, given".**

Thanks, we'll revise it.

**69. L606-608 But then how comes that prediction error is systematically lower in non-forest ecosystems than in forests?**

This comes down to the choice of metric used – they each give different information. If we consider the absolute error, prediction in non-forest is more accurate. But this is because the magnitudes are larger. A 10% error in a forest ecosystem might look huge compared to a 50% error in a semi-arid landscape. But if we consider the consistency and concordance between the pattern of predictions and observations, forest SOC is more predictable with higher $R^2$ and LCCC. We'll revise our discussion here to make this clearer in the revised manuscript.

**70. L607 change "afford" with "show".**

Thanks, we'll revise it.

**71. L610-611 and a higher C influx due to animal residues?**

Definitely. We'll add it.

**72. L627 change random forest with RF.**

Thanks, we'll revise it.

**73. L628 change "more conservative" with "lower".**

Thanks, we'll revise it.

**74. L627-628 how much?**

Sorry for the missing information, we'll add the citation of figure and table showing the results.

**75. L673-675 Is there a relationship between C input and MAP in the data?**

Yes, they're positively correlated with $R^2$ at 0.6.

**76. L691 why do the authors think that forest and grassland are explained by the same predictors, while shrubland and woodland by the same ones too? Unless this is already mentioned, please add something to the discussion.**

Sorry for the missing information. It's concluded based on the analysis of predictor importance (Figure 3) but there's a lack of text describing the results systematically. We'll revise both the Conclusion and Result to make them more consistent.

---

## Author Response (AR1)

**Responses to Referee 1**

Review of "An ensemble estimate of Australian soil organic carbon using machine learning and process-based modelling" by Wang et al. for consideration in *EGUsphere.*

Based on in situ observation data of SOC across Australia, the authors investigated the controlling factors of SOC using machine learning models and predict the gridded SOC map in Australia using trained machine learning models and process-based models. The topic of this study is interesting and important. The manuscript is well organized, and the conclusions of this study can be robustly supported by the analysis results. Nonetheless, a few more analysis and information should be added to further improve this paper.

We would like to thank the reviewers for their time and feedback on this manuscript. Please find our point-to-point responses below.

1.  **Several soil databases and studies have provided gridded SOC map over Australia, I would suggest to compare the estimates of SOC in this study to previous estimates of SOC, and provide a brief explanation on the reasons of the differences between present and previous estimates.**

Thanks for the suggestion. Yes, there are many existing gridded SOC maps over Australia, and we have compared our results with some of those most widely used (e.g., Australian baseline map (Viscarra Rossel et al., 2014)) and newly released (e.g., Walden et al., 2023; Wadoux et al., 2023) SOC maps. Generally, the estimated SOC stocks and spatial distribution in this study agree with other SOC products, and the larger uncertainty of SOC in southeast region is partly caused by the lack of observations. Please see L659-665, 677-681, and L694-709 in the revised manuscript for details.

2.  **I would suggest the authors to include the optimized values of MIMICS model in Table 1 or a new supplementary table. As the authors has conducted leave-out cross validation. The parameter values used for predicting SOC contents across Australia, and the range of each parameter obtained from the cross validation should be showed.**

Thanks for the suggestion, we have presented the optimized parameter values in Table 3 in the revised manuscript.

3.  **In Fig. 2, the locations and numbers of the observation sites belonging to each PFT are showed. I suggest to add a similar map and box plot to show the locations, total numbers of the observation sites grouped in each of the 6 clusters based on the k-means algorithm.**

Thanks, we have added a map to show the distribution of sites belonging to each environmental cluster in Figure 4 in the revised manuscript.

4.  **Did the authors tried to optimize and evaluate MIMICS model using data from all observation sites together, rather than optimize the model parameters for each PFT or cluster? What is the performance of MIMICS if it was not optimized for each PFT or cluster?**

We didn't optimize parameters using the whole dataset, and we are confident that this will result in poorer performance in MIMICS. Prior to determining our parameter optimization approach, we conducted an analysis of SOC observations, as well as climate and soil conditions across Australia. Our investigation revealed significant variations in both climate and soil characteristics across the

continent, for example, mean annual temperature ranged from 4.75 to 29.15 °C, mean annual precipitation ranged from 107.9 to 5536.7 mm, and soil clay content ranged from 3% to 59%. Additionally, SOC observations exhibited a positive skew.

Given these diverse and nuanced environmental conditions, and the low dimensionality of MIMICS relative to real-world processes, we recognized the limitations of optimizing parameters using all observations collectively. Model results are typically poor when the optimized parameters are applied to regions where the environmental conditions deviate significantly from the average conditions used in parameter optimisation.

As described in the manuscript, we instead adopt a data-driven grouping of the observations based on plant functional types and environmental conditions. This improves parameter optimization and enhances the performance of MIMICS, particularly in regions with limited observations but comparable environmental conditions.

**Specific comments:**

5. **L35: The t/ha should be changed to t ha$^{-1}$. The form of other units used in this study should also be adapted like this.**

Thanks, we have replaced t/ha with t ha$^{-1}$ in the revised manuscript.

6. **L362: What is the 'test data'?**

We randomly selected 70% of observations to train the model and the remaining 30% to validate the model. The remaining 30% are test data. We have made this explicit in the revised manuscript, please see L338-340 for details.

7. **L361-364: Why do not train the models using all observation data first? Then using the parameters trained based on all observation data to simulate SOC stocks across Australia.**

We opted for not adopting the approach suggested based on preliminary testing and our previous experience running the model. Please see our response to (4) above for details.

8. **L446: greater àlarger**

Thanks, we replaced greater with larger in the revised manuscript, please see L466 for details.

9. **L491-496: This discussion might be one-sided and give a wrong causal relationship. Many studies also suggested that SOM is one of the key controlling factors of soil bulk density (BD), as the density of SOM is smaller than soil mineral particles and stimulate the formation of aggregates and activities of insects and microbes. It is better to discuss the interaction between SOM and BD, rather than only the effect of BD on SOM.**

Yes, we agree with this comment. BD and SOM are intimately related. We added discussion on these interactions at L505-514 in the revised manuscript.

**The authors applied different models to predict the spatial variability of SOC concentration in different Australian ecosystems, using 1000 in-situ observations. The article is well written and structured and I think it should be considered for publication at EGUsphere. However, there are some flaws that should be addressed by the authors before publication.**

We would like to thank the reviewers for their time and feedback on this manuscript. Please find our point-by-point responses below.

1. **First, the authors use SOC stocks to calculate SOC concentration using bulk density from the Soil and Landscape Grid National Soil Attributes Maps (SLGA). This is counter intuitive since SOC stocks cannot be measured but are usually estimated from SOC concentration and bulk density measurements. Why the authors don't use source data for SOC concentration? Was the SOC stock from their database initially calculated using the same bulk density data that they use?**

It's correct that we used bulk density from the SLGA data, however, our explanation wasn't clear, and we apologise for that. Let us explain. We did not use direct measurements of bulk density because they were not available. The recalculation of SOC concentration from SOC stocks and bulk density was to better compare the results from the simulations. We simulated SOC flux with MIMICS in $mg/cm^3$, and one of our datasets only had SOC stocks in t/ha. Unfortunately, the lack of new soil measurements over the country means that we can really only use datasets that are currently available. To compare the simulated SOC to the observations, we converted the units of both simulated SOC and observed SOC to g C/kg soil using the SLGA-derived bulk densities at each site. These unit conversions will not affect the results of MIMICS.

To compare the performance of MIMICS and machine learning models using metrics such as RMSE and MAE which are unit-specific, we also used SOC in g C/kg soil to train the machine learning models. We will clarify this in the revised manuscript.

We rewrote the description of data to make it clearer, please see L308-316 for details.

2. **One of the main findings is that bulk density is one of the main predictors of SOC concentration. However, the common understanding is that SOC concentration is a predictor of bulk density so I think they should refine their modelling scheme to avoid biases in their result and revise the discussion on this part.**

Yes, that is correct. There is a clear (and known) negative relationship between SOC and bulk density. For example, higher bulk density usually corresponds to less porosity restricting plant root growth, oxygen availability to soil microbial communities and then SOC dynamics. Conversely, soil that is less dense has greater porosity, allows water infiltration, oxygen diffusion, root growth, microaggregate formation and enhances SOC dynamics. Therefore, it is reasonable to use bulk density as a predictor of SOC in our machine learning models.

We added the discussion on the interaction between soil bulk density and SOC in the revised manuscript, please see L505-514 for details.

3. **Second, they use a vertically discretized version of MIMICS, but then only show results for the top 30 cm. What was the rationale behind using this vertically discretized model? I think it would be interesting to show the results – at least in the supplementary material – on the SOC concentration also for deeper layers. Even if they won't be able to compare**

it with observations, if possible, they can discuss their findings in comparison to the literature.

Thanks for the suggestion. We used the vertically discretised version of MIMICS in this study mainly to consider the carbon loss from top 30 cm soil to deeper layers due to vertical transport processes. This aspect is lacking in the default version of MIMICS, where the only output of SOC is microbial respiration.

We compared the statistics between simulated SOC concentration in 30-150 cm soil by MIMICS (calculated as the mean value of 30-150 cm at 10 cm thickness) and SOC concentration extracted from a digital soil mapping product SLGA (calculated as weighted average of 30-60 cm, 60-100 cm and 100-200 cm soil). The mean value of SOC simulated by MIMICS-PFT (5.39 g C/kg soil) and MIMICS-ENV (6.54 g C/kg soil) are higher than that of SLGA (2.65 g C/kg soil) but lower than the mean value of the map representing 95% confidence interval of SLGA (6.99 g C/kg soil).

As noted by the reviewer, while we applied the vertically discretised version in this study, we only had observations for top 30 cm soil. The deeper layers were actually excluded from our parameter optimization process, they were not included in the calculation of the cost function. As SOC content in 0-30 cm is usually greater than that in deep soil layers, applying parameters optimized for 0-30 cm layer to deeper layers is likely to overestimate SOC at depth, making the predictions highly uncertain and unreliable. Therefore, we think it's less meaningful to present the results for deep layers in this paper.

**General comments**

4. **Throughout the text, it is often unclear whether they refer to SOC stocks or SOC concentration. Please add this information each time when this is appropriate.**

Thanks, we used SOC concentration to train the models, and finally estimate SOC stocks by multiplying SOC concentration by bulk density. Please see L356-362 for details. We have also specified SOC concentration or SOC stocks when it's necessary in the revised manuscript.

5. **In the discussion, they should make an effort to better compare their results to the existing literature.**

Thanks for the suggestion. We compared the performance of models used in our study with other models applied to Australia and discussed the potential reasons for the difference. Please see L583-601 and L603-613 in the revised manuscript for details.

We also compared the estimates of SOC stocks in this study with some of those most widely used (e.g., Australian baseline map (Viscarra Rossel et al., 2014)) and newly released (e.g., Walden et al., 2023; Wadoux et al., 2023) SOC maps in Australia. Generally, the estimated SOC stocks and spatial distribution in this study agree with other SOC products, and the larger uncertainty of SOC in southeast region is partly caused by the lack of observations. Please see L659-665, L677-681, and L694-709 in the revised manuscript for details.

6. **They chose to call one of their clustering groups plant functional types, but what they use are actually ecosystems. Please change this name throughout the text.**

Thanks for your suggestion regarding the classification of observations into ecosystems rather than plant functional types. However, we suggest that in this context "plant functional types" are appropriate, for two reasons,

1) The vegetation type is extracted from the National Vegetation Information Systems (NVIS) where the vegetation is divided into 32 "Major Vegetation Groups'. We aggregate them into four main types to guarantee sufficient observations for each group for model optimization. The aggregation is based solely on vegetation, without the consideration about other environmental characteristics.

2) "Plant functional types" are the classifications used in almost all process-based land models (e.g., land surface models) to represent plant and ecosystem diversity, so that using plant functional types here aligns better with existing literature and modelling approaches. This includes some existing applications of the MIMICS model (e.g. https://doi. org/10.1029/2020JG006205)

**7. The authors should add a map of the sites used and specify how many sites did they have for each ecosystem.**

Thanks for the suggestion. Figure 2 shows the spatial distribution and the number of sites in each plant functional type, and we added a similar map for each environmental group in Figure 4 in the revised manuscript.

**Specific comments**

**8. L23 I think plant functional types is not really a good definition of what you used. You used ecosystems rather than PFT.**

This is the same comment as 6 above. Please see our reply above.

**9. L28 when you say non-forest soils, you need to specify that you're not including croplands.**

Thanks, we have clarified that at L26 and L132-133 in the revised manuscript.

**10. L45-46 reference needed.**

Thanks, we have added references here (Jackson et al., 2017, https://doi.org/10.1146/annurev-ecolsys-112414-054234). Please see L44 in the revised manuscript for details.

**11. L90-91 but also the parametrization and the lack of model validation with data (see Le Noe et al, 2023)**

**Le Noë, Julia, Stefano Manzoni, Rose Abramoff, Tobias Bölscher, Elisa Bruni, Rémi Cardinael, Philippe Ciais, et al. "Soil Organic Carbon Models Need Independent Time-Series Validation for Reliable Prediction." Communications Earth & Environment 4, no. 1 (May 8, 2023): 158. https://doi.org/10.1038/s43247-023-00830-5.**

Thanks for the addition. We have revised the sentence to include this information, please see L89-92 in the revised manuscript for details.

**12. L93 change "models has rarely" with "models have rarely"**

Thanks, we have amended this, please see L94 in the revised manuscript.

**13. L91-94 Please add: and the difficulty of constraining the parameters of microbial explicit C models with relevant data on microbial properties, for example extracellular enzyme activities which are very hard to measure but also very important for these type of models.**

Thanks, we revised the sentence here to include the importance of model parameters relating to microbial activities in SOC models with explicit representation of microbes. Please see L93-97 in the revised manuscript for details.

**14. L95 change "understanding on the" with "understanding of the"**

Thanks, we have revised it at L98 in the revised manuscript.

**15. L96-98 You can also cite: Todd-Brown, K. E. O., J. T. Randerson, W. M. Post, F. M. Hoffman, C. Tarnocai, E. A. G. Schuur, and S. D. Allison. "Causes of Variation in Soil Carbon Simulations from CMIP5 Earth System Models and Comparison with Observations." *Biogeosciences* 10, no. 3 (March 13, 2013): 1717–36. https://doi.org/10.5194/bg-10-1717-2013.**

Thanks, it's really a good paper and we have cited it at L96 in the revise manuscript.

**16. L107-108 please specify what you mean with moderately well.**

Thanks, we have rewritten this sentence to make it clear,

"Wang et al. (2018a) trained boosted regression trees and random forest models using field observations and applied the trained random forest model to map the spatial distribution of SOC at two soil depths (0-5 cm and 0-30 cm) for the semi-arid rangelands of eastern Australia".

Please see L108-111 in the revised manuscript for details.

**17. L122 add "low" before amount of data.**

Thanks, we have changed this sentence to,

"However, despite the progress made in SOC modelling, significant uncertainties persist in SOC estimates due to the inherent complexities of SOC variations and the lack of appropriately sampled SOC observations."

Please see 122-124 in revised manuscript for details.

**18. L129 add in forests, woodlands, shrublands and grasslands.**

Thanks, we have demonstrate that cropland is excluded in this study here, please see L132-133 in the revised manuscript for details.

**19. L143 change The" MIcrobial-MIneral Carbon Stabilization (MIMICS)" with "MIMICS"**

Thanks, we have changed it at L146 in the revised manuscript.

**20. L144 change "soil carbon" with "SOC".**

Thanks, we reorganised the sentence here to describe the model,

"The MIMICS model (Wieder et al., 2015; Zhang et al., 2020) explicitly considers relationships between litter quality, functional trade-offs in microbial physiology, and the physical protection of microbial by-products in forming stable soil organic matter."

please see L146-148 in the revised manuscript for details.

**21. L158 change "soil carbon" with "SOC".**

Thanks, we have changed it to SOC at L160 in the revised manuscript.

**22. L180 average conditions during what period?**

We used climate data from 1991 to 2020 in this study (Table 2).

**23. L181 please specify what you mean by "a sensible range".**

It means "a reasonable range" – we have rewritten the sentence to provide the fraction of each soil carbon pool we used in MIMICS model. Please see L182-185 in the revised manuscript for details.

**24. L194-196 "We trained the RF model with different numbers of trees": how many?**

We set the number of trees as 100, 200, 300, 400 and 500, and the performance of random forest didn't vary much with different number of trees. We therefore chose a tree number of 200 in this study. We have included this at L198 in the revised manuscript.

**25. L203 use aggregate instead of segregate?**

Thanks, we change the word to "divide" to make it clearer. Please see L208 in the revised manuscript.

**26. L221 what predictors? please specify.**

Thanks, we have added "see Section 2.4" here guiding to the section describing the predictors. Please see L226 in the revised manuscript.

**27. L223 Change permutation variable importance with PVI.**

Thanks, we have revised it at L228 in the revised manuscript.

**28. L226 Change random forest with RF.**

Thanks, we have revised it at L231 in the revised manuscript.

**29. L246 in Table 1, add the units of the parameters.**

The parameters are scaling factors and are therefore unitless. We have added this information to the title of Table 1 in the revised manuscript.

**30. L250 change "The second approach was taking" with "The second approach consisted in taking".**

Thanks, we have amended this to "the second approach used the most influential abiotic variables as predictors." Please see L254 in the revised manuscript.

**31. L251 specify which were the main predictors.**

Thanks for the suggestion, the method to identify the main predictors is described in Section 2.2, and the main predictors are described in the Result (Section 3.1). As we're describing the method in this section, we think we could specify the main predictors in the Result section.

**32. L269-270 reference needed for SILO database.**

We showed the reference in Table 2. For better readability we have moved it to main text. Please see L272 in the revised manuscript for details.

**33. L275-278 In grasslands, or even in managed forests, NPP should be corrected by removing the fraction of NPP that is appropriated by human activities. Also, there might be high variabilities due to C input sources. The best would be to use different sources to see the uncertainties, but at least this should be discussed.**

Thanks for the good suggestion here. We agree that the sources of NPP data will cause uncertainties in SOC estimations, and we have demonstrated it in the revised manuscript. Please see L280-281 and L613-619 for details.

**34. L281 reference needed for NVIS.**

Sorry for the missing reference. We have added the link to download the data at L284-285 in the revised manuscript.

**35. L285 reference needed for SLGA.**

We show the reference in Table 2. For better readability we have moved it to main text. Please see L290 in the revised manuscript.

**36. L309-311 do these datasets use comparable methodologies for soil analysis?**

The methods used to measure SOC concentration in the laboratory are different because they are from different projects, see Viscarra Rossel et al., (2019) and Bissett et al., (2016). However, the methods used to process and harmonise the datasets are the same and follow the approach in Viscarra Rossel et al., (2014). Briefly, these two datasets used are harmonised to represent SOC in the 0-30 cm layer using natural cubic splines for profiles with observation at more than 3 layers and a log-log model to profiles with observation at two layers. We have added the reference describing the data harmonisation method at L312 in the revised manuscript.

**37. L312 change "algorithm" with "equation".**

Thanks, we rewrote this part to better describe the data, and this sentence was deleted.

**38. L317-318 specify the values used for S and I.**

The equation is applied to each site, so the parameters S and I are site-specific.

We added the reference describing the data harmonisation method at L312 in the revised manuscript.

**39. L 332-336 move (a) to the beginning of the sentence. same with (b). Also, you need to add (a) and (b) in the figure.**

Thanks, we have revised the figure, please see Figure 2 in the revised manuscript.

**40. L340-342 Change the sentence as follows: For machine learning models, all observations were randomly separated into a training and test datasets, with […].**

Thanks, we have revised the sentence as,

"For machine learning models, 70% of the observations were randomly selected as training data to train the models and the remaining 30% used as test data to validate the predictions of SOC concentration."

Please see L338-340 in the revised manuscript.

**41. L343 what groups?**

Each plant functional type or environmental group divided using K-means. We have clarified this at L340-341 in the revised manuscript.

**42. L340-345 And then what values did you keep after cross-validation?**

We selected the set of parameters which made MIMICS perform best on out-of-sample test data based on the metrics, please see L364-368 in the revised manuscript for details. We have also added the optimized value of each parameter used to predict SOC stocks at the continental scale in Table 3 in the revised manuscript.

**43. L370 change "Permutation Variable Importance (PVI)" with PVI.**

Thanks, we have revised it, please L373 in revised manuscript.

**44. L411 change random forest with RF.**

Thanks, we have revised it, please see L413 in the revised manuscript.

**45. L415 add "but with a higher variability across sites" at the end of the sentence.**

Thanks, we have added it at L417 in the revised manuscript.

**46. L455 change "approach" with "model".**

Thanks, we have revised it at L473 in the revised manuscript.

**47. L457-459 but what about the relative standard deviation. is it still positively correlated with the mean? it's logic that higher mean ==> higher standard deviation in absolute terms, but it would be interesting to see if also the relative standard deviation is higher.**

Thanks for the suggestion. We have replaced the map of standard deviation with a new map representing the coefficient of variation calculated as the ratio of standard deviation and mean across the four estimates. Please see L474-476 and Figure 7d in the revised manuscript.

**48. L475 In table 2, add the results for the multi-model mean.**

Thanks, we have added the results for the ensemble estimate at Table 4 in the revised manuscript.

**49. L483-484 add SOC spatial variations?**

Thanks, we revised it to "SOC spatial variation" at L496 in the revised manuscript.

**50. L488 what does it entail in terms of uncertainty the fact that the number of predictors you used was lower than that employed in most digital mapping?**

We think the uncertainty caused by the limited number of predictors is relatively small.

Most of studies that used machine learning for digital mapping of SOC typically employed different categories of predictors, including climate, organisms (e.g., plants), soil properties, terrain attributes (e.g., elevation) and land-use. In each category they usually selected several proxies, for example, temperature, precipitation, and evapotranspiration were widely used to represent climate conditions. Usually more than 20 predictors were employed in the digital mapping studies. However, some predictors contributed similarly to the predictability of SOC because of their collinearity, and some predictors contributed little because they were less important. Therefore, removing the redundant predictors and keeping only the most important ones (as assessed, for example, by random forest variable importance indicators) may actually provide not just comparable skill, but increased interpretability of SOC variations.

In our study, we used mean annual temperature, mean annual precipitation, soil clay content, bulk density, and NPP to train the machine learning model. These predictors were widely found to be important in SOC variations. Taking them as predictors, our random forest model showed good performance in predicting SOC concentration with $R^2$ greater than 0.9 for out-of-sample test data. In this case we think the predictors we used are sufficient to interpret SOC variations in Australia. While including additional predictors clearly may improve machine learning performance, the enhancement will not be large, given that $R^2$ is already greater than 0.9.

Equally important is limiting predictors to be comparable to the predictors given to the MIMICS model. This allows us to understand how well MIMICS is utilising the information available to it. Therefore, we opt to retain a limited number of predictors to facilitate a more meaningful comparison among different models.

**51. L491 is there a bias because you calculated concentration from SOC stocks and bulk density?**

We used SOC concentration (g C/kg soil) to train random forest model in order to compare models' performance based on metrics like RMSE and MAE (please see our reply to the first comment above). To test if the importance of bulk density on SOC concentration is universal, we

1) estimated variable importance using only forest SOC observations from BASE dataset where SOC is originally reported in concentration (%). Bulk density also ranked first among all predictors.
2) Estimated variable importance using a global SOC concentration dataset (WOSIS, https://www.isric.org/explore/wosis), and bulk density also ranked first in SOC concentration (unpublished result).
3) Referred to papers (e.g., https://doi.org/10.1007/s11104-015-2380-1) that also found the importance of bulk density on SOC concentration in Australia.

Therefore, we think the bias due to the calculation of SOC concentration using bulk density is small.

**52. L493-494 And land-use?**

Yes land-use significantly affects soil bulk density.

We have rewritten this part to better discuss the interaction between soil bulk density and SOC content, please see L505-514 for details.

**53. L501 change much more with much higher.**

Thanks, we have revised it at L520 in revised manuscript.

**54. L500-502 what other attributes did they use? does it mean that the number of predictors you used is not sufficient? Can you test that?**

They used around 30 predictors relating to climate, soil properties and so on. The predictors we used are sufficient in this study, please see our reply to (50) above for details.

**55. L517-518 add "in these ecosystems" at the end of the sentence.**

Thanks, we have revised the sentence as,

"MAP and NPP exhibited relatively higher influence on SOC variations in soils under these vegetation types."

Please see L536-538 in the revised manuscript.

**56. L521-523 due to what?**

Soil bulk density in forest is much lower than that in soils under other vegetation types. This is a finding based on the analysis of observations. It may be attributed to the root penetration, the formation of aggregate and stable structure due to higher content of soil organic matters in forests. And organic matters have much lower bulk density than other soil constitutes such as clay, silt and

sand, therefore, forest soils with high SOC content usually have lower bulk density than non-forest soils. On the other hand, lower bulk density improves the oxygen availability to microbial communities, accelerating their activities for carbon stabilization.

We described the reason why lower bulk density is related to higher SOC content at L505-514 and L543-546 in the revised manuscript.

**57. L538 change process-based model with process-based models.**

Thanks, we have revised it at L553 in the revised manuscript.

**58. L558 remove "two".**

Thanks, we have removed it at L579 in the revised manuscript.

**59. L562 change moderately well with moderately good.**

Thanks, we have revised it at L582 in the revised manuscript.

**60. L568 change "exit" with "exist".**

Sorry for the typo here. We have amended it at L588 in the revised manuscript.

**61. L571 change "at 30 cm soil" with "at 30 cm depth".**

Thanks, we have revised it to "for top 30 cm soil". Please see L591 in the revised manuscript.

**62. L573 add "were" between "values" and "used".**

Thanks, we have added it at L593 in the revised manuscript.

**63. L582-584 where does MIMICS get wrong?**

Based on the findings in this study, we can first contribute the lower accuracy of MIMICS to the lack of explicit representation of a soil moisture effect. MAP is important for SOC variations especially in shrublands and woodlands (Figure 3) and taking it into account for parameter optimization (MIMICS-ENV) benefits the model performance (greater $R^2$ and LCCC in Figure 6) in these vegetation types. Though MAP clearly isn't the same as soil moisture, it highlights the importance of water in SOC dynamics. Please see L621-630 in the revised manuscript for details.

MIMICS is established based on our understanding on SOC turnover processes. However, some processes are simplified or ignored because of the lack of knowledge and observational data, as well as the increased uncertainties when new parameters are introduced. Machine learning models learn patterns from the data without relying on the explicit mechanisms of SOC dynamics, therefore can better capture and handle the complex relationships between predictors and SOC.

**64. L586-588 change the sentence as follows: In their study, NPP and MAT had the highest explanatory power for SOC stocks according to MIMICS, while clay content had the highest explanatory power according to the SOC stock(???) observations, which limits the predictability of SOC using MIMICS.**

Thanks, we have revised the sentence at L607-611 in the revised manuscript.

SOC is originally reported in concentration (%), but model output is in $mg/cm^3$. To compare model outputs and observations, we need to unify the unit using bulk density. When we say SOC stock observations, we mean converted observed SOC concentration to SOC stocks using bulk density to compare with model outputs.

**65. L588-590 Do you mean according to the random forest model applied to predict SOC concentration?**

Yes, according to the PVI value showing the importance of predictors (Figure 3). PVI values are estimated using random forest (see Section 2.2).

**66. L602-603 which means that...**

SOC concentration in forest soils is much higher and more variable than that in soils under other vegetation types, which will likely lead to a higher MAE and RMSE. However, based on the higher $R^2$ and LCCC metrics for forest, we conclude that SOC in forest soil is more predictable because it aligns well with the observed patterns.

We have revised the sentences at L633-637 in the revised manuscript to make it clearer.

**67. L603-604 what do you mean? Do you mean it limits the accuracy?**

The sentence here aims to explain why the MAE and RMSE for forest SOC is higher than those for other vegetation types (please see our reply to (66) above for details). Predictions of SOC in forest likely have larger absolute errors but also larger consistency to observed patterns.

**68. L606 add "that" after "Forests, given".**

Thanks, we have revised it at L637 in the revised manuscript.

**69. L606-608 But then how comes that prediction error is systematically lower in non-forest ecosystems than in forests?**

This comes down to the choice of metric used – they each give different information. If we consider the absolute error, prediction in non-forest is more accurate. But this is because the magnitudes are larger. A 10% error in a forest ecosystem might look huge compared to a 50% error in a semi-arid landscape. But if we consider the consistency and concordance between the pattern of predictions and observations, forest SOC is more predictable with higher $R^2$ and LCCC.

We have revised it at L633-637 in the revised manuscript.

**70. L607 change "afford" with "show".**

Thanks, we have revised it at L638 in the revised manuscript.

**71. L610-611 and a higher C influx due to animal residues?**

Definitely. We have extended our discussion to better consider the effect of grazing on SOC dynamics at L640-644 in the revised manuscript.

**72. L627 change random forest with RF.**

Thanks, we have changed it at L658 in the revised manuscript.

**73. L628 change "more conservative" with "lower".**

Thanks, we have changed it at L659 in the revised manuscript.

**74. L627-628 how much?**

Sorry for the missing information, we have added the citation of Table 4 at L659 in the revised manuscript to show the results.

**75. L673-675 Is there a relationship between C input and MAP in the data?**

Yes, they're positively correlated with $R^2$ at 0.6.

**76. L691 why do the authors think that forest and grassland are explained by the same predictors, while shrubland and woodland by the same ones too? Unless this is already mentioned, please add something to the discussion.**

Sorry for the missing information. It's concluded based on the analysis of predictor importance (Figure 3) but there's a lack of text describing the results systematically. For better conclusion of the effects of different variables on predicting SOC concentrations, we have rewritten this paragraph. Please see L722-725 in the revised manuscript for details.

---

## Author Response (AR2)

Dear Editor,

We sincerely appreciate your time and feedback on our manuscript. We have updated Figure 7 to remove the repetitive patterns, and updated the link of codes in 'Code Availability' section. Please see the revised manuscript for details.

Sincerely,

Lingfei